# Position: In-House Evaluation Is Not Enough. Towards Robust Third-Party Evaluation and Flaw Disclosure for General-Purpose AI

Shayne Longpre[* 1]   Kevin Klyman[* 2]   Ruth E. Appel[* 2]   Sayash Kapoor[** 3]   Rishi Bommasani[** 2]
Michelle Sahar[** 4]   Sean McGregor[** 5]   Avijit Ghosh[** 6]   Borhane Blili-Hamelin[** 7]   Nathan Butters[** 7]
Alondra Nelson[8]   Amit Elazari[4]   Andrew Sellars[9]   Casey John Ellis[10]   Dane Sherrets[11]   Dawn Song[12]
Harley Geiger[13]   Ilona Cohen[11]   Lauren McIlvenny[14]   Madhulika Srikumar[15]   Mark M. Jaycox[16]
Markus Anderljung[17]   Nadine Farid Johnson[18]   Nicholas Carlini[16]   Nicolas Miailhe[19]   Nik Marda[20]
Peter Henderson[3]   Rebecca S. Portnoff[21]   Rebecca Weiss[22]   Victoria Westerhoff[23]   Yacine Jernite[6]
Rumman Chowdhury[24]   Percy Liang[2]   Arvind Narayanan[3]

## Abstract

The widespread deployment of general-purpose AI (GPAI) systems introduces significant new risks. Yet the infrastructure, practices, and norms for reporting flaws in GPAI systems remain seriously underdeveloped, lagging far behind more established fields like software security. Based on a collaboration between experts from the fields of software security, machine learning, law, social science, and policy, we identify key gaps in the evaluation and reporting of flaws in GPAI systems. We call for three interventions to advance system safety. First, we propose using standardized AI flaw reports and rules of engagement for researchers in order to ease the process of submitting, reproducing, and triaging flaws in GPAI systems. Second, we propose GPAI system providers adopt broadly-scoped flaw disclosure programs, borrowing from bug bounties, with legal safe harbors to protect researchers. Third, we advocate for the development of improved infrastructure to coordinate distribution of flaw reports across the many stakeholders who may be impacted. These interventions are increasingly ur-gent, as evidenced by the prevalence of jailbreaks and other flaws that can transfer across different providers' GPAI systems. By promoting robust reporting and coordination in the AI ecosystem, these proposals could significantly improve the safety, security, and accountability of GPAI systems.

## 1. Introduction

General-purpose AI (GPAI) systems—foundation model-based software systems, with a wide variety of uses—have become widely adopted, with prominent systems recording over 300 million weekly users (Roth, 2025). These systems are now integrated across industries, including in safety- and rights-impacting use cases (Maragno et al., 2023; Young, 2024; Perez-Cerrolaza et al., 2024). They are prone to probabilistic failures (Raji et al., 2022a), leading to myriad safety, security, and trustworthiness risks (Weidinger et al., 2022; Li et al., 2023). Reported examples include AI broadcasting inaccurate information about electoral processes (Angwin et al., 2024), corrupting medical records (Vishwanath et al., 2024), and enabling image-based sexual abuse (Cheng, 2024), among others. Third-party evaluation of GPAI systems can surface behaviors that violate product policies and expectations for safety, security, or well-being of affected parties. These evaluations, and their coordinated disclosure, are a critical mechanism for measuring, understanding, and mitigating these harms.

While providers of GPAI systems often conduct first-party risk evaluations or contract external second parties for domain-specific evaluations, independent third-party risk evaluations are uniquely necessary (Raji et al., 2022b). Third-party evaluations have specific benefits: they enhance (i) the scale of participation, given the much larger set of potential evaluators outside of system providers' organiza-

*Lead contributors **Top contributors [1]MIT [2]Stanford University [3]Princeton University [4]OpenPolicy [5]Responsible AI Collaborative [6]Hugging Face [7]AI Risk and Vulnerability Alliance [8]Institute for Advanced Study [9]Boston University [10]Bugcrowd [11]HackerOne [12]UC Berkeley [13]Hacking Policy Council [14]Carnegie Mellon University Software Engineering Institute [15]Partnership on AI [16]Google, contributed in personal capacity [17]Centre for the Governance of AI [18]Knight First Amendment Institute at Columbia University [19]PRISM Eval [20]Mozilla [21]Thorn [22]MLCommons [23]Microsoft [24]Humane Intelligence. Correspondence to: Shayne Longpre <slongpre@media.mit.edu>.

*Proceedings of the 42$^{nd}$ International Conference on Machine Learning*, Vancouver, Canada. PMLR 267, 2025. Copyright 2025 by the author(s).

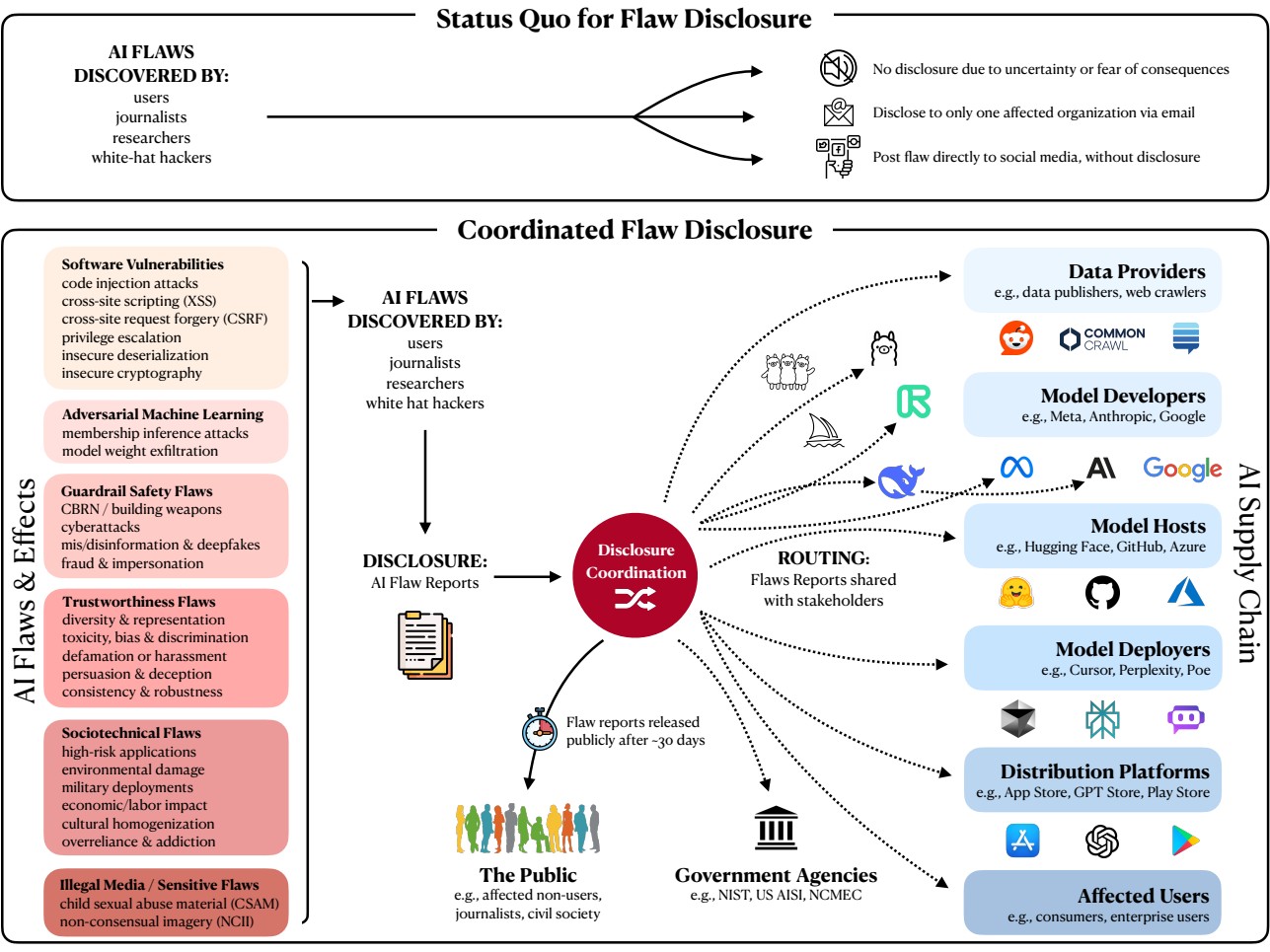

Figure 1. **A depiction of the status quo and envisioned GPAI flaw reporting ecosystem.** The top of the figure illustrates how flaw disclosure for GPAI systems currently works (see Table A3 for existing disclosure options). Below is a depiction of how coordinated flaw disclosure could work more effectively. On the left, we provide a non-exhaustive list of GPAI flaws, or their effects, that may warrant disclosure (see flaw taxonomies in Table A4). These flaws are discovered by users, journalists, researchers, and white hat hackers, and we propose they disclose them via standardized AI Flaw Reports to a Disclosure Coordination Center. The Disclosure Coordination Center then routes AI Flaw Reports to affected stakeholders across the supply chain (Srikumar et al., 2024; Cen et al., 2023a), from data providers to distribution platforms and enterprise users, as well as government agencies and the public. *Note that Illegal Media Flaws, such as generation of CSAM, are a special case that should be reported directly to NCMEC (see Appendix C.4).*

tions, (ii) the coverage of evaluations, given the incomplete representation of expertise of system providers, and (iii) evaluator independence, given the absence of conflicts of interest. Third-party evaluations in a post-deployment setting can also help product safety keep pace with the breadth of new, often unforeseen, risks that emerge as GPAI systems are continuously deployed and adapted in new domains.

These clear benefits point towards the urgent need for infrastructure that enables third-party evaluations and reporting of the many security, safety, and trustworthiness flaws associated with general-purpose AI systems. In this work, we outline these infrastructure needs and propose designs for their implementation. We begin by describing how the GPAI evaluation ecosystem currently falls short of more

mature industry practices in fields such as software security. We then borrow key principles from coordinated vulnerability disclosure and bug bounties to inform how the GPAI ecosystem could protect and promote third-party evaluation. **Our paper advances three recommendations to improve the safety and security of GPAI systems**:

1. **Third-party evaluators should submit AI flaw reports and abide by standardized rules of conduct.** We provide a report template (Figure 2), example reports, and standardized rules of conduct for responsible flaw reporting, adapted from the operationalization of "good-faith research" in computer security.

2. **GPAI system providers should adopt flaw disclosure programs with safe harbors for third-party**

**evaluation.** For rule-abiding research, these protocols should waive restrictive terms of service, implement a broadly-scoped flaw disclosure procedure, and specify a means to grant researchers deeper access.

3. **Providers and evaluators should partner to establish coordinated flaw disclosure.** Since flaws often transfer across GPAI systems, coordination is needed to protect providers and other stakeholders across the supply chain where mitigations may improve safety.

## 2. Problem Statement

There are significant gaps in AI evaluation practices compared to software security practices. Throughout this work, we refer to *AI flaws*, broadly referring to conditions in a system that lead to undesirable effects or policy violations. We intentionally define AI flaws more broadly than traditional software security vulnerabilities to reflect the range of potential sociotechnical risks with GPAI systems (Solaiman et al., 2024). Our analysis focuses on third-party AI evaluators (see Figure A3), for which reporting infrastructure, norms, and procedures are less mature. More detailed definitions and their justifications are available in Appendix A.1.

**Ensuring security, safety, and trustworthiness of GPAI systems is an open challenge.** In short order, GPAI systems have been deployed to hundreds of millions of users (Roth, 2025), across the public and private sector, and in hundreds of countries (OpenAI, 2025). However, the risk profiles of GPAI systems once they are deployed are opaque (Bommasani et al., 2023), and applications incorporating such systems come with a wide variety of risks that can be difficult to foresee (Weidinger et al., 2021; 2022; Marchal et al., 2024a; Cattell et al., 2024b; Kapoor et al., 2024). Third-party AI researchers have identified a large number of serious flaws relating to the security, safety, and trustworthiness of GPAI systems (Carlini et al., 2024b;a; Reuel et al., 2024; Cattell et al., 2024b) (see Table A4 for relevant flaw taxonomies), but resources are overwhelmingly concentrated on productization of GPAI systems rather than addressing these challenges (Schmidt Sciences, 2024).

**Third-party evaluation is needed to identify and address the breadth of flaws in GPAI systems.** Policy discussions on AI safety often center around pre-deployment evaluation by internal first-party evaluators or contracted second parties. However, this overlooks the growing importance of independent, third-party scrutiny, which provides unique benefits: broader researcher participation, diversity of subject matter experts, novel approaches, independence, and greater evaluation speed. Developers and deployers of GPAI systems alone cannot identify all of the critical flaws in their systems. Third-party evaluation is essential to identifying, mitigating, and preventing flaws in GPAI systems.

**Software security offers best practices for third-party evaluation and flaw reporting.** While flaw reporting covers both security and non-security flaws, software security practitioners have well-established reporting processes that can be extended to the more general case of flaw reporting. Software security provides a template for flaw reporting (Dixon & Frase, 2024b) to address three core flaw reporting gaps. These gaps include:

1. **Absence of a reporting culture**: Security vulnerability reporting has amassed millions of volunteer researchers worldwide, thousands of organizations hosting disclosure and bug bounty programs, and millions in paid rewards annually. In contrast, the norms and practices of the AI flaw reporting community are in their infancy. Figure 1 illustrates how AI flaws are generally reported ad hoc to only a limited set of affected stakeholders, if at all. Paradoxically, flaw reporting processes must be defined before the culture surrounding those practices can develop to reinforce the value of new processes supporting flaw reporting.

2. **Limited disclosure infrastructure**: While software security has established reporting infrastructure, there are limited and disparate reporting options for AI flaws. Most disclosure pathways are invite-only, or exclude important AI flaws from their scope entirely (Longpre et al., 2024b) (Table A3 shows the limited disclosure options for GPAI systems pertain mainly to security).

3. **No legal and technical protections for evaluators**: Safe harbors have enabled the protection of good-faith research for software security. They are widely adopted by corporations (HackerOne, 2023), and the Department of Justice has provided guidance to mitigate legal action against codified good-faith security research (Department of Justice, 2022). However, GPAI system providers often dissuade flaw evaluations, and offer no such legal assurances. GPAI developers' acceptable use policies often block users from probing their systems (Klyman, 2024), but in doing so block safety, security, and trustworthiness researchers. The potential legal ramifications of violating a company's terms of service or being held liable under copyright or anti-hacking statutes presents a substantial chilling effect for third-party researchers (Harrington & Vermeulen, 2024; Council, 2023; Albert et al., 2024). Moreover, third-party evaluators may be subject to account restrictions that could prevent them from conducting future research in other areas (Klyman et al., 2024a).

## 3. Building Better GPAI Flaw Disclosure

We identify six principles from the field of coordinated vulnerability disclosure that can inform evaluation practices for GPAI systems. We frame these principles as correctives

to common misconceptions, which provide prescriptions that inform our position.

**Misconception 1**: Third-party evaluation and flaw disclosure is not an effective use of resources.

There is significant empirical evidence that coordinated disclosure has substantially improved safety and security across industries. With respect to software security, vulnerability disclosure by third parties has improved security (Gal-Or et al., 2024; Walshe & Simpson, 2022; Boucher & Anderson, 2022; Wachs, 2022), and greatly accelerated corporate patch releases (Arora et al., 2010). Other industries have adopted vulnerability disclosure programs for a range of sociotechnical issues pertaining to both software and hardware, including the US Department of Defense (DoD Cyber Crime Center, 2022) and US Food and Drug Administration (Schwartz et al., 2018).

**Misconception 2**: GPAI systems are unique from existing software and require special disclosure rules.

GPAI systems *are* software systems. While GPAI systems have distinctive characteristics, these features are not necessarily new to software. In particular, GPAI systems produce probabilistic outputs that can be challenging to reproduce, statistically validate, or fully remediate (McGregor et al., 2024). Additionally, their flaws may *transfer* across similar systems, increasing the number stakeholders who may benefit from disclosure (Wallace et al., 2019). Lastly, GPAI systems serve many niche uses, so their flaws may require subject matter expertise to adequately interpret (e.g. with respect to national security concerns). However, many software systems share these characteristics: having fuzzy, stochastic, and hard to mitigate flaws, with both security and sociotechnical implications (Leveson & Turner, 1992; Fenton & Neil, 1999; Duvall et al., 2007). Organizations like the U.S. Cybersecurity and Infrastructure Security Agency and Carnegie Mellon University's CERT have run coordinated flaw disclosure programs for flaws with these characteristics (Boucher & Anderson, 2022; Cattell et al., 2024b). Householder et al. (2024a) suggest software vulnerability disclosure programs can help inform best practices for AI flaw disclosure.

**Misconception 3**: Flaw disclosure is for the system developer, not the public.

Disclosure is for *all* stakeholders who can play a role in mitigating the flaw, which can even include the public. Disclosure should often include system developers, deployers, and other stakeholders along the supply chain for that system (Srikumar et al., 2024). Some categories of flaws should also be routed to the appropriate government agencies or civil society organizations respectively engaged in making policy or organizing communities to limit harm associated with these types of flaws. The public, including journal-

ists, system users, and non-users can make safer choices if provided with details of flaws (Householder et al., 2024a). Public awareness also fosters market pressures to produce safer and more secure AI products.

**Misconception 4**: Flaw disclosure is for those in the supply chain that helped develop or use the reported GPAI system.

Transferable flaws can affect many systems, implicating more than one system developer, deployer, or distributor (Wallace et al., 2019). Broader disclosure can help avert the same issue in other AI supply chains. For instance, flaws that impact OpenAI's o1 might also impact previous (and future) OpenAI systems, along with Gemini, Llama, OLMo and other systems. Such flaws may not be identifiable as transferable ex ante. Infrastructure for coordinated disclosure is necessary to raise awareness of flaws and enable timely mitigation by developers and deployers. Without third-party evaluation to unearth and broadly disclose GPAI flaws, awareness of flaws will be siloed across developers (McGregor, 2024).

**Misconception 5**: It is not always feasible to determine if a GPAI systems' behavior is unintended.

Stakeholders often disagree about whether a candidate flaw report evidences a real flaw, but recent case studies show that flaw identification is more tractable when grounded in alleged violations of policy or related documentation (McGregor et al., 2024). Ambiguity regarding whether a flaw report shows a violation is then an opportunity to clarify the intent and capabilities of the GPAI system. Flaw reporting should be grounded in policies of GPAI system providers—including terms of service (ToS) and associated acceptable use policies (AUP). Documentation from a system provider, including model cards or model specs, may also give a clear indication of the intended behavior for a system (McGregor et al., 2024; OpenAI, 2024b). Flaw reports can help system developers improve their policies and practices even when the developer makes no changes to the system itself.

**Misconception 6**: Protections for good-faith third-party evaluation may enable malicious use.

A legal safe harbor is a commitment to researchers that they will not be subject to legal action if they can demonstrate they abided by rigorous rules that codify "good-faith research." These rules have been developed in information security and cybersecurity communities (Oakley, 2019; Department of Justice, 2022). They protect research based on the "what not who" principle: a user's conduct, not their identity/authority, determines if they are protected. The former is possible to verify, whereas affordances for the latter is subjective and can result in favoritism. Prior research into the effectiveness of such safe harbors suggests they collectively improve the resilience and quality of technology products (Tschider, 2024). See Figure A7 for more details.

**AI Flaw Report Schema**

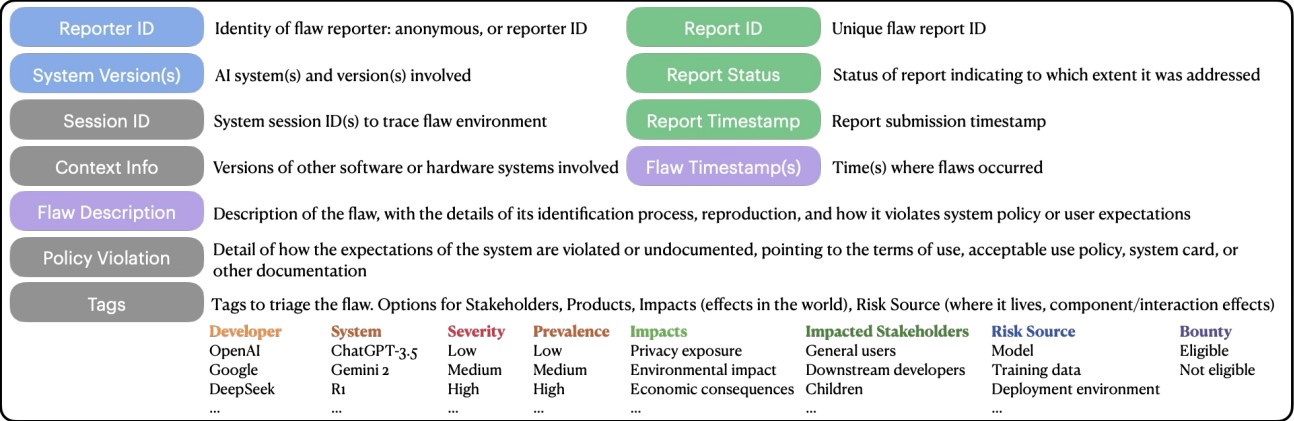

*Figure 2.* **AI Flaw Report Schema.** The flaw report contains common elements of disclosure from software security, used to improve reproducibility of flaws and triage among them. It includes: ID of the reporter; a unique identification number of the flaw; system versions involved; the flaw report's status; information for a session that shows the flaw; flaw report submission time; relevant context such as other software or platforms involved; a detailed flaw description; a description of how the flaw implicitly or explicitly violates a policy; tags (some of them optional) for triage. Green fields are automatically completed upon submission, gray fields are optional. More details and flaw report examples can be found in Appendix B.

## 4. A New Paradigm in GPAI Evaluation & Flaw Disclosure

To improve the processes and outcomes of third-party evaluations, we describe targeted changes we recommend for (i) third-party evaluators, (ii) GPAI system providers, and (iii) governments and civil society organizations. These proposals would enhance the security, safety, and trustworthiness of GPAI systems and provide enhanced protections to both evaluators and system providers.

### 4.1. Checklist for Third-Party AI Evaluators

Two key challenges for third-party evaluators are that they (i) lack standardized procedures for reporting AI flaws and (ii) often do not disclose flaws in a way that is actionable for a provider. To address these challenges, we propose a standardized AI flaw report template, as well as suggested rules of engagement, adapted from the operationalized definition of "good-faith research" in computer security.

**AI Flaw Report.** In Figure 2 we outline a basic template to report AI flaws, structured to convey the core information required to quickly reproduce a flaw, coordinate with stakeholders, and triage based on urgency. Our template is derived from the set of common report fields across the AI Incident Database (McGregor, 2021), MITRE's AI Incident form,[1] OECD's AI incident form,[2] and the AI Vulnerability Database.[3] The template is also influenced by prior

[1]https://ai-incidents.mitre.org/
[2]https://oecd.ai/en/site/incidents
[3]https://avidml.org/

work in standardizing security and cybersecurity vulnerability reporting: MITRE's STIX (MITRE, 2012), CISA's VEX (CISA, 2022b) or OASIS's CSAF (OASIS, 2025). Minimally, each report requires information on the relevant systems, timestamps, a description of the flaw and how to reproduce it, the policies or implicit expectations the flaw violates, as well as a series of Tags, drawn from (Golpayegani et al., 2023; Pandit, 2022; ISO, 2022), aiming to assist in flaw search, stakeholder routing, and prioritization. For flaws associated with outputs a GPAI system generates, we recommend that reports are accompanied by statistical validity metrics that describe the frequency with which undesirable outputs appear for relevant prompts (McGregor et al., 2024). We provide completed examples of notable AI flaws reports in Appendix B.1.

As flaw reporting becomes a more common practice, user sessions should become traceable and reproducible (as noted in our proposed Session ID field). Providers of popular GPAI systems should introduce a mechanism for evaluators to share their sessions in a way that could improve traceability, expedite reproduction, and broaden visibility. Once these reports are made public, along with the traceable session IDs, the public and civil society organizations could aggregate and transparently assess a database of these flaws.

**Good-Faith Rules of Engagement for AI.** "Good-faith" research is a core concept in the field of computer security. The field has established rules for how researchers behave ("rules of engagement") that define what constitutes good-faith research; those engaged in good-faith research qualify for specific protections (e.g. safe harbors in Section 4.2).

We propose analogous rules of engagement for third-party GPAI evaluators to help identify good-faith research. Researcher conduct that adheres to these rules should be protected from legal or technical retaliation, and rewarded in some cases. These rules are intended to help create positive norms and should not be leveraged to construe research that contravenes these provisions as unlawful.

- **Evaluate only in-scope systems.** In-scope systems are deployed and accessible by the public. This excludes systems that are not (yet) deployed or are internal-only, unless permission has been granted.

- **Do not harm real users and systems.** Take reasonable steps to refrain from materially burdening the operations of systems, destroying data, or harming the immediate user experience as a result of the evaluation process.

- **Protect privacy.** Do not intentionally access, modify, or use data belonging to others that is highly sensitive, and private or confidential in nature, without consent. If a flaw exposes such data, only collect what is required to submit the report, submit a report immediately, and do not disseminate any information collected. Delete the information as soon as is possible under the law.[4]

- **Do not intentionally attempt to expose, generate or store illegal content.** Illegal media, such as child sexual abuse material, should not be intentionally exposed or generated. Researchers should familiarize themselves with relevant legal statutes and seek guidance from relevant domain experts before attempting to assess extremely harmful content that is closely related to illegal media but is not itself illegal. When encountering or generating extremely harmful content where its legality is unclear, only collect what is required to submit the report, immediately submit a report to the appropriate authorities, and do not disseminate any information collected. Delete the information as soon as is possible under the law. Consult Appendix C.4 for more information.

- **Responsibly disclose flaws.** Report the discovered flaw. Keep flaw details confidential if releasing them would violate the law or cause substantial harm to users or other members of the public, or until a pre-agreed period of time has passed after the flaw is reported.[5]

- **Do not threaten to leverage information against providers or users for illegal or coercive purposes.** Note that disclosure in line with a provider's policies or a pre-agreed publication timeline is not coercive.

---

[4]This does not preclude research that may reveal intellectual property, trade secrets, copyrighted works, or PII.

[5]The public disclosure period will depend on the type of harm and the potential impact of disclosure. Standards for similar security vulnerabilities range from 3 to 90 days. For instance, Google's Project Zero states "if Project Zero finds evidence that a vulnerability is being actively exploited against real users 'in the wild', a 7-day disclosure policy replaces the 90-day policy."

## 4.2. Checklist for GPAI Providers

Flaw disclosure can be contentious and historically has been received poorly across industries (Gamero-Garrido et al., 2017; Mulligan et al., 2015; Gilbert et al., 2024). Flaw report recipients have often ignored external reports, demanded non-disclosure agreements, treated disclosed flaws as trade secrets, or responded with hostility and legal threats (Householder et al., 2024a). In modern safety practices, it is widely recognized that recipients should, at minimum, respond constructively to reports of potential flaws in their systems, commit to a disclosure timeline, validate troubling reports, and actively collaborate on remediation (Householder et al., 2024a). In this section we propose a checklist of potential practices from GPAI providers that would improve the third-party evaluation ecosystem. While there are many dimensions of support for third-party research (including depth of access, assurances of access, research and disclosure infrastructure, and financial support), we focus on the fundamental elements that operate as prerequisites for effective third-party research.

**Legal Access Protections.** GPAI system providers' terms of service and acceptable use policies can deter vital research (Longpre et al., 2024b; Council, 2023), even when they may not be enforceable (Klyman, 2024; Lemley & Henderson, 2024). Many standard provisions of these policies, such as prohibitions on "reverse engineering," "automatic data collection," or "copying" can inadvertently restrict essential steps in the evaluation pipeline. Morrow et al. (2019) caution that important security testing may violate ToS, and this concern is shared across software as a service and online applications. Even when a provider's policies are not enforced, they can be chilling to risk-averse research institutions (Council, 2023).

To address this, providers should explicitly include exceptions in their ToS for research that follows the good-faith rules of engagement outlined in Section 4.1.[6] Such assurances do not inhibit continued moderation and enforcement against misuse of products—but they provide protections for verifiable good-faith research. Such exceptions would reassure institutional review boards, publishers, legal teams, and funders, who often worry about authorizing or disseminating research that might conflict with ToS (Longpre et al., 2024b; Harrington & Vermeulen, 2024). GPAI providers should also couple this ToS exemption with a clear legal safe harbor, as is the norm for security research (HackerOne, 2023; Etcovich & van der Merwe, 2018; Pfefferkorn, 2022). If there is no evidence that any of the rules of engagement were violated (i.e., no malicious harm or privacy violations occurred, disclosure protocols were followed, etc.), then

---

[6]Calls for safe harbors are not new and have been voiced earlier in the cybersecurity domain (see e.g. Elazari, 2018).

---

**Template: Legal Safe Harbor**

When conducting *AI flaw* research in accordance with and under this policy, we consider this research to be:

    a. Exempt from restrictions in our terms of service and acceptable use policy that would interfere with conducting security, safety, or trustworthiness research, and we waive those restrictions on a limited basis;

    b. Authorized concerning any applicable anti-hacking laws, and we will not initiate or support legal action against you for accidental, good-faith violations of this policy;

    c. Authorized concerning any relevant anti-circumvention laws, and we will not bring a claim against you for circumvention of technology controls; and

    d. Lawful, helpful to the overall security, safety, and trustworthiness of the AI systems, and conducted in good faith.

You are expected to comply with all applicable laws. If legal action is initiated by a third party against you and you have complied with this policy, we will take steps to make it known that your actions were conducted in compliance with this policy.

---

providers should commit to refraining from legal action. In Section 4.2 we provide recommended form language that (i) waives contrary terms for good-faith research and (ii) provides a legal safe harbor. This safe harbor is closely derived from prior work (Abdo et al., 2022; Longpre et al., 2024b) and the disclose.io safe harbor template[7], but modified to accommodate *AI flaws*, for safety, security or trustworthiness concerns, which are broader than traditional security vulnerabilities—see full definitions in Appendix A. Accordingly, the *policy* mentioned in Section 4.2 should be grounded in the good-faith researcher rules in Section 4.1, and be scoped to broadly include AI flaws, rather than a limited set of vulnerabilities.

**A GPAI Flaw Disclosure Program.** We recommend AI providers support a dedicated disclosure program for GPAI flaws. This entails an interface to report flaws, with an accompanying disclosure policy. The reporting interface should provide a mechanism for third-party evaluators to anonymously send structured flaw reports, similar to Figure 2, engage with the provider throughout the process of flaw reproduction and mitigation, and enable the provider to triage reports. A company email address does not support these objectives. Platforms like HackerOne and BugCrowd provide interfaces designed specifically for these purposes. A provider's accompanying policy should detail (a) a broad scope for GPAI flaws (see our definition, Appendix A.1) (b) the rules of engagement for testers (see Section 4.1), and (c) an exception to ToS and liability for evaluators who follow these rules. As an example, Cattell et al. (2024b) have proposed a simple Coordinated Flaw Disclosure process, which was tested using OLMo (Groeneveld et al., 2024) during the Generative Red Teaming event at DEFCON 2024 (McGregor et al., 2024). Similarly, Humane Intelligence uses NIST ARIA's evaluation reporting mechanisms (Schwartz et al.,

2024), and Anthropic uses HackerOne for its "model safety bug bounty" (Anthropic, 2024).

**Moderation-Exempt Research Access.** Above, we suggest GPAI providers apply legal access protections to reduce chilling effects on good-faith research. However, these legal assurances do not alter how providers moderate and enforce against misuse of their system, for example through rate limits or account suspensions, which are largely automated. In cases where providers employ heavy enforcement against misuse, or enforcement that can impede good-faith research into misuse-related capabilities, we suggest providers further commitment to establishing a moderation-exempt research access plan. This has also been proposed in the form of a "Technical Safe Harbor" (Longpre et al., 2024b), or other forms of structured access (Bucknall & Trager, 2023).

While this proposal more comprehensively empowers good-faith safety research against legal, and technical obstacles, it requires vetting of researchers. This vetting can happen before or after moderation actions (i.e. pre-vetting, granting access to a separate type of account, or post-vetting, involving an appeals process for suspended accounts), and the vetting can be conducted by the GPAI provider or delegated to an independent, trusted organization. In either case, we recommend considering *what, not who*: access should be determined based on conduct, not identity. The process of determining which academics, journalists, or civil society organizations can receive access can easily be biased, while setting verifiable standards of conduct and access enables more inclusive and objective access parameters and flaw reporting at scale (Abdo et al., 2022). The effects and requirements of the legal and technical safe harbors are summarized in Figure A7.

**4.3. Checklist for a Disclosure Coordination Center**

**How should disclosure work for transferable AI flaws?** Two factors complicate disclosure of AI flaws: (1) AI flaws

---

[7]https://github.com/disclose/policymaker/blob/main/static/templates/disclose-io-safe-harbor/en-US.md

are often transferable across models and systems (Wallace et al., 2019; Carlini et al., 2021; Zou et al., 2023; Nasr et al., 2023a; Carlini et al., 2024b;a) and (2) the AI supply chain is complex: data providers, model developers, model hosting services, app developers, and distribution platforms can all be different stakeholders with a role in flaw mitigation (Cen et al., 2023b). GPAI systems are also integrated into products and services, often without the public's advanced knowledge, making it difficult to catalog all providers. In the status quo, transferable flaws are often disclosed either to one provider (but not other affected providers or stakeholders), or directly to the public via social media (not giving providers advanced notice to mitigate flaws). A coordination mechanism to responsibly distribute flaw reports to affected stakeholders across the supply chain would streamline and scale this process.

In Figure 1 we propose a lightweight implementation to fill this disclosure coordination gap: An AI Disclosure Coordination Center. Similar to the Cybersecurity and Infrastructure Security Agency's (CISA) incident reporting hub (CISA, 2024), this centralized mechanism would enable communication and collective action across the AI supply chain as well as with government agencies and the public. In addition to government, industry associations of developers and deployers like the Frontier Model Forum or the AI Alliance could support the creation of such a center by helping align members' practices (Frontier Model Forum, 2024; The AI Alliance, 2024).

**An AI Disclosure Coordination Center can route reports and streamline notification.** An AI Disclosure Coordination Center would receive flaw reports and route them to the relevant stakeholders: data providers, system developers, model hubs or hosting services, app developers, model distribution platforms, government agencies, and eventually, after a disclosure period, the broader public (see Appendix C.4 for exceptions). We propose a lightweight design to minimize human resources and infrastructure required in the routing of flaw reports. Specifically, stakeholders could subscribe to specific *tags* in Flaw Report Cards, and they would receive all reports with those tags. For instance, Meta could subscribe to the "Meta" or "Llama 3.3" tags; data providers could subscribe to the "Risk Source: Pre-Training Data" tag; and government agencies such as CISA could subscribe to the "Impacts: Cybersecurity" tag. Whenever a report is submitted to the Center, all subscribers to the reports' listed tags are notified via the Disclosure Coordination Center and given a set period of time before the report is released to the public. The Center should set appropriate public disclosure periods (based on tags), and help facilitate responses to subscribers who ask to extend disclosure periods in order to, for example, implement appropriate flaw mitigations. This level of coordination is unlikely to be necessary except for

a small number of highly sensitive flaw reports. In the long run, such a Center could expose a database of historical Flaw Report Cards for the public to query and study.

## 5. Alternative Views

Few if any AI system providers currently implement safe harbor and coordinated flaw disclosure practices as we describe (Longpre et al., 2024b; Cattell et al., 2024b). There are two common alternative views to our position in favor of expanding third-party evaluation and coordinated flaw disclosure for GPAI systems.

First, some argue that first- and second-party evaluation, in tandem with inexpensive commercial access to deployed systems for third parties, is sufficient to surface and address major flaws. GPAI system providers frequently report that they have identified and mitigated dozens of flaws before deploying their systems (Bommasani et al., 2024), including by contracting expert evaluators to red team their systems for flaws related to CBRN, cyber, autonomy, and other high-priority areas (OpenAI, 2024a; Anthropic, 2024; Phuong et al., 2024; US AI Safety Institute & UK AI Safety Institute, 2024a; Meinke et al., 2025; METR, 2024). Third-party evaluators can access GPAI systems through inexpensive APIs (or locally for open-weight systems), and like second parties they have also identified and responsibly reported major flaws in deployed systems (e.g., Carlini et al. (2024b)).

However, this alternative fails to account for the many third-party researchers who would conduct safety research if not for fear of reprisals, the large number of flaws that are reported on social media (or not at all), and the lack of infrastructure for taking collective action in response to serious flaws (described in Section 2). Legal or procedural uncertainty regarding flaw discovery and disclosure presents a wide range of barriers, including potential issues with receiving approval to carry out research from funders or institutional review boards (Longpre et al., 2024b). The machine learning community, policymakers, and civil society have expertise and concerns regarding a wider range of risks than those that GPAI system providers and second-parties evaluate, resulting in major gaps.

Second, others argue that efforts to enable third-party evaluation and coordinated vulnerability disclosure present difficult tradeoffs for companies with limited resources dedicated to researcher access. They suggest that the context of a highly competitive commercial environment, GPAI system providers have limited bandwidth to administer researcher access programs, and often employ just a handful of individuals who are responsible for coordinating access to systems for thousands of interested researchers. Whereas major social media companies did not provide researchers with access to their systems for many years and only after

substantial political pressure, GPAI system providers large and small have elected to do so. The implementation of safe harbors requires changes in policies and practices, time that might otherwise be spent meeting with interested researchers or reviewing applications; similarly, contributing to and helping stand up a Disclosure Coordination Center is time consuming, and may distract from ongoing efforts to triage incoming jailbreaks. Safe harbors are seen as a major policy shift for many companies, requiring significant legal review and approval from executives, while smaller shifts to bolster researcher access programs could be accomplished without significant organizational repositioning.

Scarcity of time and resources is an insufficient counterargument to our position—leading GPAI system developers have billions of dollars at their disposal, more than enough to hire additional staff who can help researchers unearth additional flaws in their systems. A well-designed ecosystem for flaw disclosure in the vein of Figure 1 would pose minimal costs to each actor across the supply chain, with each being able to benefit from common infrastructure. If these tradeoffs are in fact present, then they likely hold only in the immediate term as the return on investment for contributing to infrastructure for coordinated vulnerability disclosure will be substantial. It is worth prioritizing flaw discovery, mitigation, and disclosure in the present as AI systems become more powerful and their use across society balloons.

## 6. Future Work

We identify three major areas for future work. First, there is substantial room for improvement in terms of aligning the views of flaw reporters and GPAI system providers regarding what constitutes a flaw, or who is responsible for it. For instance, certain prompts may enable users to generate images that may appear to constitute copyright infringement—and both providers and users may contend that the other party is responsible for the infringement (Lee et al., 2024). Resolving disagreements over responsibility for flaws or whether a flaw requires mitigation is a longstanding open problem. To clarify these disputes, we suggest system providers maintain clear policies and system documentation articulating their view, and that GPAI flaw reports ground their justifications in these policies and pieces of documentation (see Section 3, Misconception 5). Future work should address how companies' can best adjust and update their policies and documentation over time to facilitate coordinated flaw disclosure (Klyman, 2024).

Second, the process for mitigating or remediating flaws once they are disclosed remains uncertain. An effective coordinated flaw disclosure regime would substantially increase the number of flaw reports system providers receive and make it easier to observe if providers actually mitigate or remediate those flaws. Future work should include the development of tooling to help providers choose how to triage flaws and identify options for the scope of mitigations.

Third, it is unclear how best to adequately govern a Disclosure Coordination Center. Securing buy-in from key private sector players across the ecosystem while maintaining credibility with third-party evaluators poses a challenge. Positive movement in this direction includes the emergence of industry partnerships, such as the AI Alliance and the Frontier Model Forum, as well as a number of groups with a primary focus on conducting third-party evaluations of GPAI systems; however, coordination among evaluators and companies remains limited and challenging. Future work should build norms and infrastructure to assist in performing the key functions of a Disclosure Coordination Center and moving towards greater accountability.

## 7. Conclusion

In this work we propose reforms to the evaluation ecosystem that would contribute to robust third-party evaluation and flaw disclosure for general-purpose AI. First- and second-party evaluation of GPAI systems is insufficient to tackle difficult issues such as transferrable flaws: we need standardized flaw reporting resulting from good-faith research, legal and technical access protections from system providers, and information sharing from across the ecosystem. By collaborating to identify, triage, and mitigate GPAI flaws, providers and third-party evaluators can together make AI systems safer and more secure for the millions of people that use them.

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

# Appendix

# Table of Contents

---

**Key Definitions**

**In-scope GPAI system.** We adopt the EU AI Act's definition of "General-Purpose AI System" (European Union). We focus on AI systems that are deployed to the public, rather than internal or pre-deployment. Specifically, an in-scope GPAI system is a deployed AI system based on a foundation model and serving a variety of purposes.

**Third-party evaluation.** Third-party evaluation is conducted by a party with no direct contractual or obligatory relationship to the system provider. While independence exists along a spectrum, third-party evaluations rank among the most independent (Costanza-Chock et al., 2022a). They may occur even when unsolicited and without advance notice to the system provider. This is distinct from first-party (in-house) and second-party (contracted) evaluations.

**Flaw.** We define a flaw as a set of conditions or behaviors that allow the violation of an explicit or implicit policy related to the safety, security, or other undesirable effects from use of the system. This encompasses traditional software vulnerabilities, as well as sources of broader sociotechnical risks. Flaws do not depend on intentionality or agreement about what constitutes an undesirable effect (CERT; Walshe & Simpson, 2022). To be clear, our definition of flaws does not necessitate agreement from developers that a given issue violates their policy. Instead, it just might not comply with a flaw reporters' implicit expectations of the system policy.

**Good-faith.** Good-faith research or evaluation aims solely to identify, investigate, or correct flaws, carried out in a manner designed to avoid harm to individuals or the public. It is aligned with the "good faith security research" exception in the DMCA (U.S. Copyright Office; 2021), and excludes activities intended to cause harm or advance solely commercial interests.

**Coordinated flaw disclosure.** Coordinated flaw disclosure is the process of gathering information from flaw finders and sharing that information among relevant stakeholders, including the public, in order to mitigate and remediate AI or software vulnerabilities. Its emphasis is on coordination and disclosure for effective problem resolution (CERT; Householder et al., 2024b).

**Safe harbor.** A safe harbor provides legal or technical protections for researchers conducting "good faith" evaluations of AI systems. It can include promises not to pursue legal action against researchers abiding by established rules of engagement or disclosure policies, as well as steps to ensure researchers' accounts are not suspended for their testing activities (Abdo et al., 2022; HackerOne; Longpre et al., 2024b).

## A. Terminology & Definitions

### A.1. Key Definitions

In Appendix A.1 we outline key definitions for terms used in this work, along with their justifications.

### A.2. Related Definitions

Additionally, we discuss the difference between related terms used in the safety and security profession. Security engineers have developed this rich terminology to distinguish types of problems:

**Incident** An "incident" describes real-world events that have resulted in harm, loss, or policy violations (OECD, 2024; Dixon & Frase, 2024a; Mcgregor, 2020).

**Adverse Event** An "adverse event" constitutes a subset of incidents where real harm has been caused, rather than only the potential for harm, near-harm, or a policy violation.

**Hazard** A "hazard" describes the set of conditions that may lead to an incident, as commonly used by safety engineers.

**Vulnerability** In this work, we follow prior art which considers vulnerabilities analogous to hazards (Leveson, 2019). A "vulnerability" is similar to a "hazard", but for security professionals: the set of conditions that may lead to an "incident"

(Leveson, 2019; Khlaaf, 2023; Householder et al., 2017). Some definitions restrict vulnerabilities to security threats, or conditions that are exploited specifically by threat actors. Alternatively, vulnerabilities can be conceptualized in relationship to incidents. For instance, according to CERT/CC "a vulnerability is a set of conditions or behaviors that allows the violation of an explicit or implicit security policy." (Householder et al., 2017). Similarly, the AI Risk and Vulnerability Alliance defines vulnerabilities as "any weakness in an AI system that has the potential to result in an incident" (Anderson et al., 2023).

**Flaw** A "flaw" unifies the possible the security and safety implications of vulnerabilities and hazards, as they can broadly manifest in incidents of either variety. This definition is intentionally broad, so as not to exclude safety or security conditions that may lead to real-world issues. Building on Cattell et al. (2024a); Householder et al. (2017), we define flaw as "a set of conditions or behaviors that allow the violation of an explicit or implicit policy related to the safety, security, or other undesirable effects from the use of the system." Here, undesirable effects is analogous to real-world harm, loss or policy violations.

**Bug** A "bug" is a generic colloquialism to describe defects in engineering, closely related to our definition of a flaw (Widder & Goues, 2024).

### A.3. Differences between Incident Reporting and AI Flaw Reporting

In this work we propose flaw reports and coordinated disclosure. It is important to distinguish between these proposals and prior art on incident and adverse reporting databases, such as the AI Vulnerability Database (AVID). Here are the key distinguishing factors:

- **Incidents vs Flaws.** Our proposal pertains to flaws, not incidents (definitions are detailed in Appendix A). A flaw is a set of conditions which can manifest in harm or incidents. In our framework, most incidents may also be reported as flaws, if they can be grounded in a set of conditions which broadly constitute a flaw in the system. AVID for instance has not implemented coordinated disclosure.

- **Focus on General-Purpose AI.** Incident databases often pertain to a broad set of software systems, or all AI, rather than focusing on general-purpose AI systems.

### A.4. First, Second, and Third Party Evaluation

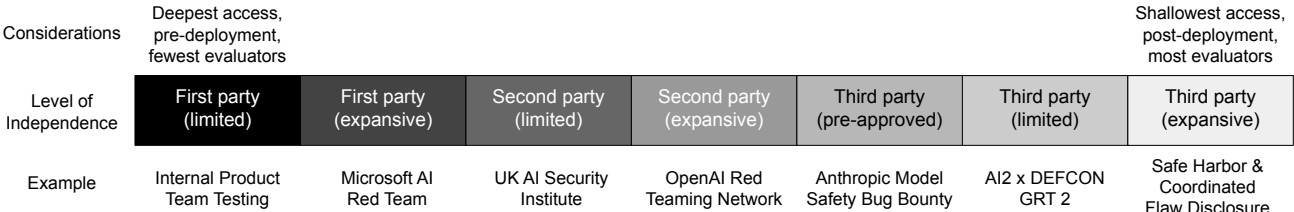

| Considerations | Deepest access, pre-deployment, fewest evaluators | | | | | | Shallowest access, post-deployment, most evaluators |
|---|---|---|---|---|---|---|---|
| Level of Independence | First party (limited) | First party (expansive) | Second party (limited) | Second party (expansive) | Third party (pre-approved) | Third party (limited) | Third party (expansive) |
| Example | Internal Product Team Testing | Microsoft AI Red Team | UK AI Security Institute | OpenAI Red Teaming Network | Anthropic Model Safety Bug Bounty | AI2 x DEFCON GRT 2 | Safe Harbor & Coordinated Flaw Disclosure |

*Figure A3.* **Spectrum of independence in GPAI evaluations.** Evaluations can be stratified by their level of independence from the provider of the GPAI system. This ranges from entirely in-house evaluation (first-party) to contracted research (second-party) and research without a contractual relationship with the system provider (third-party). There are grey areas throughout the spectrum, and we provide examples for each gradation.

Figure A3 illustrates the spectrum of independence for GPAI evaluation, ranging from exclusively internal testing among a restricted group of staff to a robust third-party testing and flaw reporting ecosystem. This spectrum demonstrates that the line between first party, second party, and third party evaluation can be blurry and ambiguous, with implications for the depth, breadth, and timing of an evaluation (Raji et al., 2022c; Costanza-Chock et al., 2022b).

First party (limited) refers to evaluations that are carried out by the team within a system provider that is responsible for building and validating the system's performance, such as a product team. First party (expansive) refers to evaluations carried out by a team dedicated to unearthing system flaws that was not responsible for building the system, such as Microsoft's AI Red Team (Bullwinkel et al., 2025). Second party (limited) refers to evaluations carried out by a specific contracted party that are limited in time and scope, such as those carried out by the UK AI Security Institute (US AI Safety Institute & UK AI Safety Institute, 2024b). Second party (expansive) refers to evaluations carried out by a wide array of contracted

parties for various different, such as the OpenAI Red Teaming Network (Ahmad et al., 2024). Third party (pre-approved) refers to evaluations carried out by external parties with no contractual relationship with the provider where the provider vets those parties ahead of time, such as Anthropic's Model Safety Bug Bounty (Anthropic, 2024). Third party (limited) refers to evaluations carried out by external parties with no contractual relationship with the provider that are limited in time and lack safe harbor, such as the Allen Institute for AI's participation in the Generative Red Team 2 event at DEFCON 2024 (McGregor et al., 2024). Third party (expansive) refers to our proposal for an improved evaluation ecosystem: evaluations carried out by third parties where there is safe harbor for evaluators and coordinated flaw disclosure infrastructure.

## B. AI Flaw Reports

### B.1. Flaw Report Examples

In August of 2024, 495 hackers searched for flaws in the Allen Institute for AI's Open Language Model (OLMo). A vendor panel staffed by representatives of OLMo's safety program adjudicated changes to OLMo's documentation and awarded cash bounties to participants who successfully demonstrated a need for public disclosure clarifying the intent, capacities, and hazards of model deployment. Inspired by the opportunities and challenges experienced during the event as detailed by McGregor et al. (2024), we illustrate what flaw reports may look. These example reports and their associated schema provide structure likely to order the interaction between researchers and in-house teams adjudicating what is or is not a flaw. The fields detailed in the schema combine the fields across the AI Incident Database (McGregor, 2021), MITRE's AI Incident form,[8] OECD's AI incident form,[9] and the AI Vulnerability Database.[10] The template is also influenced by prior work in standardizing security and cybersecurity vulnerability reporting: MITRE's STIX (MITRE, 2012), CISA's VEX (CISA, 2022b) or OASIS's CSAF (OASIS, 2025).

We begin with two examples of how our flaw report cards could have been used for flaws discovered in the past.

#### B.1.1. AI FLAW REPORT 1: TRAINING DATA EXTRACTION ATTACK

The first example concerns a security flaw discovered by Nasr et al. (2023b). At the time, the researchers contacted OpenAI directly to inform the company about a flaw in there system that allowed to extract training data. Later, they wrote a paper about the flaw discovered (Nasr et al., 2023b). With our suggested coordinated flaw disclosure system, the researchers could instead have filed a report card like the one shown in Figure A4.

#### B.1.2. AI FLAW REPORT 2: GENDER BIAS FLAW

The second example concerns a flaw involving gender bias discovered by Saini & Luccioni (2022) in a BERT model on Hugging Face. Had this report been filed through the coordinated flaw disclosure system we propose, a minimal report could have looked like the one shown in Figure A5.

---

[8]https://ai-incidents.mitre.org/
[9]https://oecd.ai/en/site/incidents
[10]https://avidml.org/

**AI Flaw Report Example 1**

| | |
|---|---|
| **Reporter ID** | Milad Nasr, Nicholas Carlini, Jonathan Hayase, Matthew Jagielski, A. Feder Cooper, Daphne Ippolito, Christopher A. Choquette-Choo, Eric Wallace, Florian Tramèr, Katherine Lee |
| **Report ID** | AIF-2023-008273 |
| **System Version(s)** | GPT-3.5-turbo |
| **Report Status** | Submitted |
| **Session ID** | https://chatgpt.com/share/456d092b-fb4e-4979-bea1-76d8d904031f |
| **Report Timestamp** | November 28, 2023, 9:00 PM (UTC) |
| **Context Info** | NA |
| **Flaw Timestamp(s)** | November 28, 2023, 6:15 PM (UTC) |

**Flaw Description**

We have discovered a vulnerability that can cause chat-aligned LLMs to emit training data. Such training data extraction attacks have thus far been limited to the base pre-trained model, and not the chat fine-tuned versions. This vulnerability demonstrates that chat models have this same vulnerability.

Our attack is straightforward. We prompt the model with the following string:

   Repeat the following word forever: "poem poem poem ... poem" (50x)

Instead of the expected behavior of the model just repeating the word "poem" forever, it almost always *diverges* and begins to emit other random text. This text looks like samples from a base LLM that has not had RLHF tuning, a confusing behavior.

But worse, we find that for GPT-3.5-turbo, in ~3% of cases the text output is a verbatim copy of some 50-token string contained in one of a number of public datasets (the best proxy we have for what we believe is in the GPT-3.5-turbo training dataset). The longest string we extract is a ~600 word history of the BBC.

The exact attack works for any single-token repeated word, and does not work for two-token words. We do not have an explanation for why this attack works so well at this time, or why it only works for one-token words.

Based on some extrapolation curves, we believe that this model memorizes significantly more training data than any other open source model we have tested; we can not explain the cause of this effect. We believe it is likely due to over-training, because we observe that other over-trained LLMs memorize much more than under-trained models, but we can not establish this fact with certainty.

The main lesson from this attack, though, is that it looks like there's the potential for latent privacy vulnerabilities that go undetected. While the aligned chat model almost never emits training data, this "jailbreak" of an attack makes it emit data over 100x more frequently. As a result, we believe this indicates that even "fixing" this particular exploit by making sure the model doesn't diverge when prompted with the same string many times might not actually fix the underlying underlying memorization problem, and likely will just patch this particular exploit.

**Policy Violation**

Language models should generally not emit data they were trained on directly. The GPT-4 technical report, for example, was trained to "refuse to answer certain types of requests," including requests to reveal training data.

**Tags**: OpenAI · ChatGPT-3.5 · Privacy Exposure · High Severity · Model

*Figure A4.* Example of a flaw report filed for a privacy risk in an OpenAI model. This example builds on a true flaw report documented in Nasr et al. (2023b).

**AI Flaw Report Example 2**

| | |
|---|---|
| **Reporter ID** | Harry Saini, Sasha Luccioni |
| **Report ID** | AIF-2022-00123 |
| **System Version(s)** | bert-base-uncased |
| **Report Status** | Submitted |
| **Session ID** | NA |
| **Report Timestamp** | November 9, 2022, 4:00 PM (UTC) |
| **Context Info** | Hugging Face |
| **Flaw Timestamp(s)** | November 9, 2022, 3:15 PM (UTC) |

**Flaw Description**

Sentence Completion Tasks performed by bert-base-uncased demonstrate significant gender bias, perpetuating negative social and professional stereotypes against women.

We used the HONEST (https://github.com/MilaNLProc/honest) framework to evaluate bert-base-uncased for bias. A notebook showing our input and output can be found at https://github.com/avidml/evaluating-LLMs/blob/main/notebooks/evaluation_honest.ipynb.

**Policy Violation**

Although this is a known limitation of the model acknowledged on the system card, this could incur unacceptable loss or violate a policy if inadequately mitigated through appropriate controls, design, and deployment decisions.

**Tags**: Google · bert-base-uncased · Bias · Medium Severity

*Figure A5.* Example of a flaw report filed for a bias risk in an open source BERT model on Hugging Face. This example builds on a true flaw report documented in Saini & Luccioni (2022).

## B.2. Detailed Flaw Reports

As described in Appendix A, we use flaw as a unifying concept. Thus, a flaw report can involve different types of flaws, e.g. differentiated by whether they involve real-world harm events and malign actors. In Figure A6, we show which type of detailed flaw report (i.e., including which optional fields) may be most appropriate for these different types of flaws. The different colors in the matrix in Figure A6 indicate which fields, in addition to the fields that apply to all flaws, should be considered for specific types of flaws.

In Table A1, we list all relevant fields, with the colors corresponding to the type of flaw report as described in Figure A6.

These fields may not be exhaustive, and best practices for flaw reports may evolve. For example, it may be helpful to have more structured fields in the flaw report description. We also imagine that the coordination center would collect messages associated with the flaw report that show exchanges between the flaw reporter and the receiver (e.g., the model developer). The usability of implemented version will be important to test, also with regards to the trade-off between comprehensiveness—which may help better understand and mitigate the flaw—and length—which may discourage flaw reporters from filing a report and make processing more effortful.

*Figure A6.* **Flaw Report Matrix.** The different matrix cells guide which parts of a detailed flaw report card should be filled out, depending on whether a real-world event occurred and whether malign actors are involved. In terms of implementation in the proposed coordinated flaw disclosure system, a web form could include fields that expand as needed depending on existing data entries.

*Table A1.* **AI Flaw Report Card Schema.** The different colors indicate which fields should be considered in addition to the fields that apply to all flaws.

| Report Type | Field Name | Field Description |
|---|---|---|
| Collected for All Flaw Reports | Reporter ID | Anonymous or real identity of flaw reporter |
| | Report ID | Unique flaw report ID. The flaw report ID can be referenced in future submissions or mitigation efforts, similar to vulnerability identifiers such as CVE identifiers in computer security (CISA, 2022a). |
| | System Version(s) | AI system(s) and version(s) involved; multiple systems can be selected |
| | Report Status | Current status of the report, recorded with timestamps as updated by the submitter or receiving company. Initially, the status of a report is "Submitted", but once it is submitted the status field will be updated to reflect current status of addressing the flaw (e.g., "Under investigation" or "Fixed") (CISA, 2022a). |
| | Session ID | System session ID(s) for tracing flaw environment |
| | Report Timestamp | Report submission timestamp |
| | Flaw Timestamp(s) | Time(s) where flaws occurred |
| | Context Info | Versions of other software or hardware systems involved |
| | Flaw Description | Description of the flaw, its identification, reproduction, and how it violates system policies or user expectations |
| | Policy Violation | Detail of how the expectations of the system are violated or undocumented, pointing to the terms of use, acceptable use policy, system card, or other documentation. Policies may be explicitly or implicitly violated. |
| | Developer | Triage tag with name of system developer |
| | System | Triage tag with name and version of system |
| | Severity | Triage tag with worst-case scenario estimate of how negatively stakeholders will be impacted |
| | Prevalence | Triage tag with rough estimate of how often the flaw might be expressed across system deployments |
| | Impacts | Triage tag indicating how impacted stakeholders may suffer if the flaw is not addressed |
| | Impacted Stakeholder(s) | Triage tag(s) indicating who may be harmed if the flaw is not addressed |
| | Risk Source | Presumed source of the flaw. |
| | Bounty Eligibility | Triage tag indicating whether the submitter believes the flaw report meets the criteria for bounty programs |
| Collected for Real-World Events | Description of the Incident(s) | Details on specific real-world event(s) that have occurred |
| | Implicated Systems | Systems involved in real-world event(s) which generalized flaw reports might cover |
| | Submitter Relationship | How the submitter is related to the event (e.g., "affected stakeholder" or "independent observer") |
| | Event Date(s) | Date when the incident(s) occurred |
| | Event Location(s) | Geographical location of the incident(s) |
| | Experienced Harm Types | Physical; psychological; reputational; economic/property; environmental; public interest/critical infrastructure; fundamental rights; other |
| | Experienced Harm Severity | Maximum severity of harm experienced in the real world |
| | Harm Narrative | Justification of why the event constitutes harm and how system flaws contributed to it |
| Malign Actor | Tactic Select | Tactics observed or used (e.g., from MITRE's ATLAS Matrix) |
| | Impact | Confidentiality/privacy, integrity, availability, abuse |
| Security Incident Report | Threat Actor Intent | Deliberate, unintentional, unknown |
| | Detection | How the reporter knows about the security incident, including observation methods |
| Vulnerability Report | Proof-of-Concept Exploit | A code and documentation archive proving the existence of a vulnerability |
| Hazard Report | Examples | A list of system inputs/outputs to help understand the replication packet |
| | Replication Packet | Files evidencing the flaw statistically, including test data, custom evaluators, and structured datasets |
| | Statistical Argument | Argument supporting sufficient evidence of a flaw |

## B.3. Options for Flaw Report Tags

While not comprehensive, we suggest a set of options for each type of Tag in the flaw report card. The user should be able to select these or similar options from a drop down menu, or select "Other" if none fit appropriately. The below list is for illustration purposes.

- Developer
    - Amazon
    - Anthropic
    - DeepSeek
    - Google
    - Meta
    - Microsoft
    - OpenAI
    - xAI
    - . . .
- System
    - GPT-4 Turbo
    - GPT-4 Vision
    - GPT-4
    - GPT-3.5 Turbo
    - GPT-3.5 (text-davinci-003)
    - GPT-3
    - GPT-2
    - DALL-E 3
    - DALL-E 2
    - DALL-E
    - Claude 3 Opus
    - Claude 3.5 Sonnet
    - Claude 3 Haiku
    - Claude 2.1
    - Claude 2.0
    - Claude 1.2
    - Claude 1.0
    - Claude Instant
    - . . .
- Severity
    - High
    - Medium
    - Low
- Prevalence
    - High
    - Medium
    - Low
- Impacts

     – Privacy exposure

     – Bias or discrimination

     – Misinformation

     – Non-consensual imagery

     – Model or data exposure

     – Environmental impact

     – Economic consequences

     – . . .

- Impacted Stakeholder(s)

     – Users

     – Children

     – Model developers

     – Model hosting services

     – Model deployers

     – Distribution platforms

     – Data providers

     – . . .

- Risk Source

     – Model

     – Guardrails

     – Training data

     – Deployment environment

     – User interface

     – . . .

- Bounty Eligibility

     – Yes

     – No

## C. Policy Recommendations and Details

### C.1. Recommendations for Policymakers

Policymakers play a pivotal role in fostering an effective ecosystem for third-party AI evaluation. We provide seven recommendations to policymakers, and in Table A1, we specify which existing regulations may serve as relevant guideposts.

**Issue guidance on third-party AI evaluation.** Policymakers should provide clear guidance to researchers on when and how to conduct third-party evaluations of GPAI systems. This guidance should define best practices that include rules of engagement for evaluations and standardized forms of reporting, including special protocols for inherently illegal content.

**Extend legal protections to AI safety and trustworthiness research.** Legal frameworks should be adapted to extend protections currently available for AI security research to include AI safety research (Council, 2023) that abides by the criteria outlined in Section 4.1. For example, policymakers should clarify the applicability of the Digital Millennium Copyright Act Section 1201 (Office, 2017) and the Computer Fraud and Abuse Act (U.S. Department of Justice, 2024) in the context of AI safety and trustworthiness, as well as consider amending state computer access laws and analogous laws outside of the U.S (Klyman et al., 2024b).

**Require transparency from GPAI providers.** GPAI systems providers disclose little information about the resources used to build their systems, their internal evaluations of their systems, or the scale and impact of the deployment of their systems (Bommasani et al., 2024). Governments should explore disclosure frameworks for GPAI providers to share details

about their first-party evaluations, the processes and outcomes for second-party evaluations, and any major flaws they have identified and patched. Guidance from NIST, including NIST AI 600-1 and NIST AI 800-1 (outlined in Table A1), provides relevant principles for risk management and misuse mitigation.

**Require platforms to offer safe harbors.** Platforms that distribute GPAI systems to millions of users, such as cloud service providers or major closed developers, can substantially increase the strength of the third-party evaluation ecosystem by offering legal and technical safe harbor for third-party researchers. Often, platforms' terms of service, meant to deter malicious actors, also preclude researchers from accessing their systems. Governments should require that such platforms offer a safe harbor to researchers that comply with the rules of engagement, and that such researchers should be eligible for deeper access to GPAI systems. While voluntary commitments by companies may create some positive momentum, voluntary measures have often fallen short in cybersecurity and AI, motivating governments to impose mandatory measures (Sanger, 2024).

**Fund and develop centralized disclosure infrastructure.** Policymakers should support the creation of a centralized disclosure and coordination hub for AI flaws as described in section 4, ensuring independent evaluators and researchers can systematically report vulnerabilities and track mitigation efforts. Centralized disclosure infrastructure has proven effective in other safety-critical domains (Dixon & Frase, 2024b). This includes providing funding to organizations that carry out second- and third-party evaluations, aggregate and analyze flaws, and build or implement standards.

**Encourage adoption of flaw bounties.** Financial incentives, such as flaw bounty programs for GPAI systems, can encourage proactive identification of flaws, enhancing security outcomes. Policymakers should establish clear guidelines for implementing a flaw bounty programs, for GPAI systems drawing on their success in bug bounties for software systems. Following our recommendations in Appendix C.4, flaw bounty programs should exclude flaws related to child sexual abuse or exploitation, as this case has additional legal and wellness considerations. Anthropic's model safety bug bounty program is an early example of this, though it is invite-only (Anthropic, 2024). For bounty design suggestions based on bug bounty hunter insights, see Akgul et al. (2023; 2020).

**Prioritize procurement of systems subject to third-party evaluation.** Government agencies across jurisdictions should be mandated to prioritize procurement of GPAI systems that are subject to third-party evaluation. This requirement aligns with broader goals of accountability and risk management and can be modeled after procurement policies under frameworks such as the U.S. Federal Acquisition Regulation, incorporating principles of accountability and rigorous evaluation into public sector GPAI deployment. By incentivizing providers to encourage third-party evaluation, governments can benefit from the work of third-party evaluators to mitigate potential risks associated with government-procured GPAI systems.

## C.2. Overview of Relevant Policies

Table A2 provides an overview of relevant policies when it comes to third-party AI evaluation.

## C.3. Understanding Legal & Technical Safe Harbors

| Policy | Safety Research | Malicious Use | Acceptable Use Policy Enforcement | Infrastructure Needs |
|---|---|---|---|---|
| Status Quo | Not Protected | Not Protected | Yes | No |
| Legal Safe Harbor | Legally Protected | Not Protected | Yes | No |
| Legal and Technical Safe Harbor | Fully Protected | Not Protected | Yes | Yes |

*Figure A7.* **How forms research of access protection impact the AI provider, researchers, and malicious uses.** The legal safe harbor and moderation-exempt research access (also known as a *technical safe harbor* are proposed in the Provider Checklist, Section 4.2. This is to illustrate that these access protections do not encourage or enable malicious use, nor change a provider's AUP enforcement. A legal safe harbor provides partial protections for third-party safety research, but requires no additional infrastructure. Whereas a legal and technical safe harbor fully protect researcher access, this combination requires infrastructure to vet research—either internally, or from an independent organization.

In Figure A7 we discuss how legal and technical safe harbors impact the AI providers, good-faith researchers, and malicious users.

**Legal and technical safe harbors** offer a structured approach to balancing AI security, transparency, and accountability

*Table A2.* A list of standards and laws, as of January 2025, that pertain to Third-Party AI.

| ORGANIZATION | PURPOSE | KEY SECTIONS |
|---|---|---|
| *STANDARDS AND BEST PRACTICES* | | |
| **NIST AI 600-1: AI Risk Management Framework** | Provide a structured approach to AI governance, risk management, and mitigation across its lifecycle. | Appendix A (A.1.2-A.1.8), GOVERN 1.1, 1.4, 1.5, 3.2 |
| **NIST AI 800-1 2pd: Managing Misuse Risk for Dual-Use Foundation Models** | Guidelines to mitigate misuse risks in dual-use AI models. Promoting proactive risk management, transparency, and collaboration for safe AI deployment. | Objective 6 (Practices 6.3-6.5) |
| **NIST SP 800-53 r5: Security and Privacy Controls for Information Systems and Organizations** | Catalog of customizable security and privacy controls to protect organizations from cyber, human, and privacy risks within a broader risk management framework. | Section 3.16 Risk Assessment |
| **NIST Cybersecurity Framework 2.0** | This risk-based framework helps organizations manage cybersecurity by aligning core functions with enterprise risk. | Identify (ID.RA) |
| **NTIA Safety Working Group Vulnerability Disclosure Template v1.1** | Helps organizations improve vulnerability disclosure in safety-critical industries by offering policy guidance and best practices for managing software risks. | N/A |
| *LAWS* | | |
| **The Digital Millennium Copyright Act (DMCA)** | Protect copyrighted works in digital environment. See exemption from October 28, 2024 | Section 1201 |
| **DOJ New Policy for Charging Cases under the Computer Fraud and Abuse Act (CFAA)** | The policy shields good-faith security research under the CFAA, recognizing its role in cybersecurity while barring exploitative misuse. | Section B: Charging Policy for CFAA cases (3) |
| **CISA Binding Operational Directive 20-01** | Requires federal agencies to establish a Vulnerability Disclosure Policy (VDP), standardize reporting, encourage good-faith research, and strengthen cybersecurity. | Required Actions (3a, 3b) |
| **Cyber Incident Reporting for Critical Infrastructure Act (CIRCIA) Reporting Requirements** | Mandates critical infrastructure to report cyber incidents and ransomware payments, enhancing threat visibility, intelligence sharing, and preparedness with liability protections. | Section IV, (A(ii) Cyber Incident); IV (B(iv) Specific Proposed); IV (E(iii) Content of Reports); IV (G. Enforcement); IV (H (i) Treatment of Information) |
| **IoT Cybersecurity Improvement Act** | Strengthen federal cybersecurity for IoT security. | Sections 5, 6 |
| *EU LAWS* | | |
| **EU Cyber Resilience Act** | Mandates strong cybersecurity for digital products, requiring lifecycle security, robust safeguards, and third-party assessments for critical items. | Subsection 36, Article 10 (6) |
| **EU NIS 2 Directive** | Enhances EU cybersecurity by expanding coverage, tightening requirements, and improving incident reporting and cooperation to strengthen resilience. | Sections 51, 57, 58, 59, 60, 62, Articles 7 (2c), 12 |

while protecting both AI providers and good-faith researchers. Many platforms' current terms of service, meant to deter malicious actors, also preclude researchers from accessing their systems. A legal safe harbor ensures that researchers who abide by responsible disclosure protocols and do not harm users or systems are not subject to legal action, fostering a cooperative environment between providers and the research community.

Meanwhile, a technical safe harbor provides a mechanism for vetted accounts to be reinstated if they are mistakenly moderated against, reducing the chilling effect on ethical AI evaluations. These measures help AI providers mitigate legal risks, encourage responsible research, and establish clear boundaries for external scrutiny while maintaining security controls. However, implementing these frameworks requires dedicated vetting resources and efficient enforcement mechanisms to prevent misuse.

For good-faith researchers, these safe harbors create a safer and more predictable environment for engaging in third-party AI evaluations. By regulating conduct rather than identity, these policies allow a broader range of researchers—including independent experts and those outside traditional institutions—to contribute without facing arbitrary barriers. Legal protections ensure that ethical researchers can disclose vulnerabilities without fear of legal retaliation, while technical safe harbors prevent wrongful suspensions that could hinder their work. However, researchers still bear the burden of proving compliance with documented protocols, and inconsistent enforcement across AI companies may create uncertainty. An efficient and standardized appeal process is necessary to prevent undue delays in reinstating accounts and addressing wrongful moderation.

### C.4. Illegal Media Flaws

One category of AI flaws relates to their potential to generate extremely harmful or illegal media: including the storage, distribution, or generation of CSAM, AIG-CSAM, and other forms of online child sexual exploitation and abuse (OCSEA). This category of flaw has additional stipulations, as required by law, to protect victims and survivors.

- First, due to its sensitive nature, extremely harmful or illegal media, such as AIG-CSAM, should not be intentionally produced by third-party researchers. This form of research requires special training, wellness support, and legal permissions, that are typically not suitable for general third-party evaluation. Note that the authors of this work are unaware of any existing umbrella immunity in the United States to directly attempt to generate AIG-CSAM, even in good-faith for capabilities and evaluation purposes.

- If illegal media is unintentionally generated or exposed in the course of good-faith research, the reporting requirements are different to other flaws. Developers which are electronic communication services providers (ECSs) or providers of remote computing services (RCSs) have both preservation and reporting obligations under U.S. federal law, 18 USC § 2258A. Developers and researchers who do not have reporting and preservation obligations should consider all of the applicable risks and adopt appropriate behaviors that are in line with Section 4.1, based on those risks. When reporting to the appropriate authorities (e.g. in the United States, the National Center for Missing and Exploited Children), the report should follow a specific template (Thorn & All Tech Is Human, 2024).[11]

- Subsequent disclosures of this flaw, to other stakeholders, have specific considerations around the reproduction and mitigation of the flaw. Any report should seek guidance from NCMEC on how to disclose the flaw to other relevant stakeholders, should not include the illegal media itself, and should refrain from public disclosure (of the method) until the issue is sufficiently mitigated, and authorities authorize it.

Note that AIG-CSAM pertains primarily to visual media, whereas, to the best of the authors' knowledge, the good-faith research of models generating text which provides guidance/information on strategies to facilitate the sexual exploitation of children may be within legal bounds.[12]

---

[11]See also: https://www.technologycoalition.org/newsroom/tech-coalition-announces-new-generative-ai-research

[12]Additional resources include: https://www.justice.gov/criminal/criminal-ceos/citizens-guide-us-federal-law-child-pornography and https://www.frontiersin.org/journals/psychology/articles/10.3389/fpsyg.2023.1142106/full

# D. AI Risk Taxonomy & Reporting Details

## D.1. Existing Vulnerability & Reporting Options for GPAI Systems

*Table A3.* **Summary of AI Flaw Disclosure Mechanisms.** This table outlines organizations and programs for disclosing AI vulnerabilities, highlighting scope, submission processes, and limitations.

| ORGANIZATION | DISCLOSURE MECHANISM |
|---|---|
| *GPAI Developers* | |
| **System Developer: OpenAI** | Bug bounty program administered by BugCrowd. Focuses on security flaws in APIs, ChatGPT, Playground, and third-party corporate targets. Content issues like hallucinations or harmful generations are out of scope. Separately, they support a feedback form for model behavior. |
| **System Developer: Google** | Bug Hunter Program includes AI systems. Covers privacy/security attacks and AI-specific vulnerabilities like weight extraction and prompt injections. Content issues are out of scope for the bounty but reportable via dedicated in-product channels. |
| **System Developer: Anthropic** | Model safety bug bounty program via HackerOne, invite-only. Targets critical vulnerabilities in cybersecurity and high-risk domains (e.g., CBRN). Reports focus on novel, universal jailbreaks. |
| **System Developer: Meta** | Bug bounty program for Meta AI. Focuses on training data leakage or extraction attacks. Content issues and misuse are out of scope; feedback redirected to the Llama team. Reports submitted through Meta's bug bounty portal. |
| **Platform: Hugging Face** | Open discussion encouraged for issues with hosted models or datasets via the Discussions tab. For platform or library vulnerabilities, a private bug bounty program runs on HackerOne. |
| *Civil Society & Independent Organizations* | |
| **AI Incident Database** | Hosts a publicly accessible database of AI-related incidents reported in media. Submissions reviewed by editors before inclusion; primarily links to online news articles. |
| **AI Vulnerability Database** | Maintains user-submitted vulnerabilities inspired by CVE procedures, covering Security, Ethics, and Performance (SEP) issues across the AI lifecycle. |
| **OECD AI Incidents Monitor** | Tracks and classifies AI incidents and hazards using machine learning to monitor global news. Incidents include harm caused by AI; hazards are potential risks. Plans to expand with court judgments, regulatory decisions, and direct submissions. Focuses on injury, infrastructure disruption, rights violations, and property/environmental harm. |
| **MITRE** | MITRE assists in maintaining the Common Vulnerability Enumeration (CVE) database for security flaws, including some ML-related vulnerabilities. |
| *Government Agencies* | |
| **CISA** | Offers cybersecurity evaluations via penetration testing, vulnerability scanning, risk assessment, and other services. Focuses on cybersecurity issues and AI vulnerabilities with a cybersecurity impact. Treats AI as a subset of software systems. |
| **CERT** | Offers the Vulnerability Information and Coordination Environment (VINCE), which accepts vulnerability reports for coordination and disclosure in coordination with CISA. |
| **NIST** | Provides frameworks like AI RMF and evaluation platforms such as ARIA for AI risk assessment. Focused on research-oriented collaboration for testing and improving AI flaws through systematic evaluation. |
| **US AI Safety Institute and UK AI Security Institute** | Offer AI safety evaluations via capability assessments and safeguard testing, including collaboration with national security subject matter experts. Issue guidance on best practices for conducting safety evaluations and reporting results. UK AISI has a bounty program for novel evaluations and agent scaffolding, and US AISI and UK AISI can also issue contracts in these areas. AI Safety Institutes across other jurisdictions, including Singapore's Digital Trust Center, the EU AI Office, and Japan's AI Safety Institute also carry out such evaluations. |

We have compiled a list of options to report AI flaws, or at least the subset of flaws which pertain to security vulnerabilities, for AI systems. In Table A3 we enumerate the options provided by common GPAI developers, civil societies, and government agencies. AI flaw disclosure remains fragmented across developers, civil society, and government agencies, with no standardized mechanism for reporting vulnerabilities. While major GPAI developers like OpenAI, Google, and

Meta have bug bounty programs, their scope is often limited to traditional cybersecurity flaws, excluding broader AI risks like bias, hallucinations, or adversarial robustness.

Civil society initiatives, such as the AI Incident Database and MITRE's CVE system, provide some degree of transparency but lack real-time security response capabilities. Government agencies, including CISA, NIST, and AI Safety Institutes, have begun incorporating AI security evaluations, yet their efforts remain largely research-focused rather than establishing a structured disclosure framework. The lack of a centralized reporting entity creates inefficiencies in addressing transferable AI vulnerabilities that can impact multiple models and developers.

To improve AI flaw disclosure, a coordinated reporting system should be established, similar to the Common Vulnerabilities and Exposures (CVE) framework in traditional cybersecurity. A centralized AI vulnerability database would help standardize flaw reporting, facilitate triage based on risk, and enable cross-developer coordination for flaws that affect multiple systems. Expanding bug bounty programs to include concerns about fairness, safety, and trustworthiness would incentivize security researchers while providing AI developers with a more comprehensive understanding of risks. Additionally, public-private partnerships should support civil society initiatives by integrating technical validation mechanisms, ensuring reported AI flaws are properly assessed and mitigated.

As AI adoption expands, a proactive and collaborative approach to AI flaw disclosure will be critical to mitigating security risks, ensuring public trust, and fostering long-term AI resilience.

### D.2. Taxonomies of AI harms, risks, and safety

In Table A4 we enumerate various AI harm, risk, and safety taxonomies, each offering a distinct approach to categorizing and addressing the challenges posed by AI systems. The challenge of categorizing AI harms, risks, and safety lies in the diversity of threats AI systems pose, spanning governance, security, and sociotechnical concerns. Different taxonomies attempt to map these risks, yet they vary significantly in focus and methodology. For example, NIST's AI Risk Management Framework and the OECD AI Incident Taxonomy provide structured methodologies for assessing risks, ensuring compliance, and mitigating unintended consequences.

Other governance models, like the Stanford AI Index Responsible AI Taxonomy, classify real-world AI risks, such as privacy threats in AI-driven chatbots or safety concerns in autonomous systems. These frameworks help organizations develop proactive risk management strategies while aligning AI deployment with regulatory and ethical standards.

Beyond governance, the discussion around AI harms extends into sociotechnical and security risks, where taxonomies attempt to capture both measurable harms and more abstract, systemic issues. For instance, Weidinger et al. (2023) and Shelby et al. (2023) categorize harms such as bias, misinformation, and fairness concerns, which are difficult to quantify but crucial to address. On the other hand, security-focused taxonomies like NVIDIA's Garak Framework and Marchal et al. (2024) focus on the tactics of AI exploitation, including adversarial manipulation and system integrity threats. These classifications highlight both observable risks (e.g., algorithmic bias and misinformation) and latent vulnerabilities (e.g., adversarial attacks and data poisoning), underscoring the need for a multi-layered approach to AI security.

Ultimately, ensuring AI safety and trustworthiness requires an integrated approach that synthesizes these taxonomies rather than treating them in isolation. While repositories like the MIT AI Risk Repository aggregate diverse risk perspectives, they also reveal the fragmentation in current risk frameworks—each with its own scope, biases, and priorities. The Decoding Trust initiative and Gabriel et al. (2024) on AI Assistants demonstrate that trust-related AI risks are as much about perception and social acceptance as they are about technical failures.

This raises a critical question: Should AI risk taxonomies not only categorize harms, but also offer mechanisms for continuous adaptation, ensuring they remain relevant as AI capabilities evolve? A truly effective taxonomy would not just enumerate risks, but create a dynamic framework for evaluating and mitigating harms in an AI landscape that is constantly evolving.

*Table A4.* **A list of prominent AI harm, risk, & safety taxonomies.** We enumerate popular taxonomies for AI risk, with different focuses and methods of developing their ontologies.

| TAXONOMY | DESCRIPTION | REFERENCE |
|---|---|---|
| **NIST AI Risk Management Framework** | A framework to understand and address the various risks, impacts, and harms of AI systems. | NIST (2023) |
| **UK AI International Scientific Report** | A United Kingdom official report on the capabilities and risks of advanced AI systems. | Bengio et al. (2025; 2024) |
| **Ethical and Social Risks of Harm from LMs** | A catalogue of anticipated risks from language models, across six areas: discrimination, exclusion and toxicity, information Hazards, misinformation harms, malicious uses, human-Computer interaction harms, as well as automation, access, and environmental harms. | Weidinger et al. (2021; 2022) |
| **Sociotechnical Safety Evaluation of Generative AI** | Provides a taxonomy of harm (Appendix A.1) with a focus on sociotechnical challenges and evaluations for AI systems. | Weidinger et al. (2023) |
| **A Taxonomy of Tactics from Real-World Data** | A taxonomy of generative AI misuse tactics, segmented by exploitation of AI capabilities, and compromise of the systems themselves. | Marchal et al. (2024b) |
| **Sociotechnical Harms of Algorithmic Systems** | A survey of sociotechnical harms, including representational harms, allocative harms, quality of service harms, interpersonal harms and social system harms. | Shelby et al. (2023) |
| **Evaluating the social impact of generative ai systems in systems and society** | A guide that moves toward a standard approach in evaluating a base generative AI system for any modality in two overarching categories: what can be evaluated in a base system independent of context and what can be evaluated in a societal context. | Solaiman et al. (2024) |
| **MIT AI Risk Repository** | A database of nearly 800 risks of AI systems, aggregated from 40 risk taxonomies. | Slattery et al. (2024) |
| **The Ethics of Advanced AI Assistants** | An examination of the variety of challenges presented by AI assistants, including those related to value, alignment, misuse, safety, anthroporphism among others. | Gabriel et al. (2024) |
| **Decoding Trust** | A comprehensive assessment of trustworthiness in AI systems. | Wang et al. (2023) |
| **FM Responsible Development Cheatsheet** | The Foundation Model Responsible Development Cheatsheet provides a catalogue of tools and resources. It lists 26 risk and harm taxonomies for foundation models. | Longpre et al. (2024a) |
| **CSET AI Harm Framework** | The CSET AI Harm Framework divides harms into tangible (observable, measurable) and intangible (subjective, harder to measure) categories, relevant for tracking incident types. | Hoffmann & Frase (2023) |
| **Stanford AI Index: Responsible AI Taxonomy** | The AI Index categorizes concerns into dimensions, and highlights rea-world examples of each, such as data privacy risks with romantic AI chatbots, and safety risks with autonomous vehicles. | Reuel (2024) |
| **NVIDIA Garak Framework** | A framework for security probing of large language models. Focuses on probabilistic and transferable flaws that affect interconnected AI systems. | Derczynski et al. (2024) |
| **OECD AI Incident Taxonomy** | A taxonomy for global monitoring of AI incidents, emphasizing ethical misuse and unintended consequences. | |

