# OpenReview forum: "Position: In-House Evaluation Is Not Enough. Towards Robust Third-Party Evaluation and Flaw Disclosure for General-Purpose AI"
_ICML.cc/2025/Position_Paper_Track — ICML 2025 Position Paper Track spotlightposter_

### Official Review · Reviewer_LLxu · 2025-03-07

**Significance:** 3
**Argument Clarity:** 3
**Rating:** 3
**Confidence:** 4

**Questions:**

Could the authors provide more detailed examples of successful third-party evaluation and disclosure practices in other fields, such as software security, and how these could be adapted to AI systems?

How do the authors plan to address the potential legal and ethical challenges associated with third-party evaluation of AI systems, particularly in cases where the evaluation might involve accessing or using proprietary data?

**Discussion Potential:**

3

**Paper Summary:**

The paper "Position: In-House Evaluation Is Not Enough. Towards Robust Third-Party Evaluation & Flaw Disclosure for General-Purpose AI" argues for the importance of independent third-party evaluations and coordinated disclosure mechanisms for general-purpose AI (GPAI) systems. The paper highlights the limitations of current internal and contractual evaluation practices and advocates for standardized defect reporting, legal protections for researchers, and improved infrastructure for coordinating defect disclosure. The contributions include a framework for standardized AI flaw reports, a checklist for GPAI providers to adopt flaw disclosure programs, and a proposal for a centralized AI Disclosure Coordination Center to streamline and scale the disclosure process.

**Position:**

Yes

**Position In Title:**

Yes

**Related Work:**

2

**Strengths And Weaknesses:**

The paper presents a clear and well-supported position on the need for third-party evaluation and flaw disclosure in GPAI systems. It provides a comprehensive analysis of the current gaps in evaluation practices and offers specific recommendations for improvement. The paper is well-structured, with a clear introduction, problem statement, proposed solutions, and policy recommendations. However, the paper could benefit from more detailed case studies or examples of successful third-party evaluation and disclosure practices in other fields, such as software security. Additionally, while the paper mentions several relevant works, it could further strengthen its argument by more extensively citing and discussing related research in the area of AI safety and third-party evaluation [1,2,3,4].

[1] "Coordinated Disclosure for AI: Beyond Security Vulnerabilities" by Sven Cattell, Avijit Ghosh, and Lucie-Aimée Kaffee, which discusses the challenges of applying coordinated disclosure practices to AI systems.

[2]"Outsider Oversight: Designing a Third Party Audit Ecosystem for AI Governance" by Inioluwa Deborah Raji and colleagues, which explores how third-party audit systems can enable independent oversight of AI systems.

[3]"A Safe Harbor for AI Researchers: Promoting Safety and Trustworthiness Through Good-Faith Research" from the Federation of American Scientists, which addresses the need for legal protection for independent researchers.

[4]"Concrete Problems in AI Safety" by Amodei and others, which identifies specific safety problems in current AI systems.

**Support:**

3

---

> ### Author Rebuttal · Authors · 2025-03-29
>
> We would like to thank Reviewer LLxu for their thoughtful and constructive feedback! We are pleased they found our position clear and well-supported. We put significant effort into identifying gaps in the current ecosystem, and are encouraged they found our recommendations specific and compelling.
>
> Reviewer LLxu asks us to (a) detail more examples/case studies that disclosure practices have aided software security, and (b) to more extensively cite/discuss some related research. We propose the following additions to address these recommendations.
>
> **Detail studies where disclosure practices have aided software security**
>
> See our response to reviewer nbqs, who also had this suggestion. We will supplement the paper with details on those case studies.
>
> **Cite and discuss related research**
>
> We propose adding text to further contextualize some of the most relevant citations (including the ones you point to). Note that the first 3 you listed are already cited in our paper (under “Cattell et al. 2024b”, “Raji et al., 2022b”, and “Longpre et al., 2024b” respectively). In the spirit of your feedback, we discuss key citations more extensively:
>
> Our work builds out and extends several prior proposals. First, “Coordinated Disclosure for AI” presents the motivations for AI flaw disclosure. Our work builds on this by proposing specific mechanisms for flaw reporting and coordination, as well as standardized rules of engagement, and the legal research protections that could foster better norms. Our work also extends “A Safe Harbor for AI Researchers”, which makes an initial case for legal and technical researchers protections. “To Err is AI” presents a case study of AI2 at DEFCON of nascent flaw reporting in action, minus broader coordination. It provides both grounding and empirical evidence for the design and efficacy of our reporting schema. Similarly, “Concrete Problems in AI Safety” presents a proposal for how researchers can contribute to scalable oversight and work on adversarial robustness—of which our work explores a particular solution. Lastly, “Outsider Oversight” provides a broader perspective of third-party audits, and where algorithmic bug bounties may play a role in AI responsibility and accountability.
>
>
> **Question: How would the authors address legal and ethical challenges with third-party evaluation, such as accessing proprietary data?**
>
> This is an excellent question, and one we worked extensively with security and legal experts on, in drafting the “Good-faith Rules of Engagement” in Section 4.1. Essentially, there need to be rules to protect AI developers from malicious attackers, and protect good-faith researchers who want to find (and help fix) flaws that may yield proprietary data. The rules of engagement we propose draw heavily on prior work in the software security field (see literature and real examples [1,2,3]), and are designed to conservatively separate these two cases as simply as possible. They carve out access to proprietary data as long as: the unauthorized access is reported if it yields private information; that private information is quickly deleted without dissemination; and the researcher can demonstrate they took reasonable measures to avoid harming the system or real users in the purpose of testing. There will always be cases on the margin that require legal or technical debate, but these guidelines have evolved from the security literature, in response to complex security testing requirements.
>
> [1] Longpre et al. (2024), Position: A Safe Harbor for AI Evaluation and Red Teaming. https://arxiv.org/pdf/2403.04893, ICML 2024
>
> [2] https://www.justice.gov/archives/opa/pr/department-justice-announces-new-policy-charging-cases-under-computer-fraud-and-abuse-act
>
> [3] https://bugcrowd.com/openai

---

### Official Review · Reviewer_Scur · 2025-03-11

**Significance:** 3
**Argument Clarity:** 4
**Rating:** 4
**Confidence:** 4

**Questions:**

1) What do you foresee as the outcome of this work?
2) What would be the path to adoption of this work? The current computer security infrastructure of bug bounties and the like has a legal and corporate history, that lead to the current state of systems. Would you intend for following a similar pathway, or is your idea more to piggyback extant systems, structures, and people to bring AI into that fold?
3) Would there be any negative consequences of corporatising and incentivising AI security research?

**Discussion Potential:**

2

**Paper Summary:**

This paper presents a framework for trying to move AI threat/risk management to a more traditional computer-security approach, with third party verification, incentives, and legal safe harbors.

### Post Rebuttal Summary ###

After engaging with the authors comments, I increased the score for this paper to a 4, contingent upon the authors completing clarifying works - which is something they have committed to.

**Position:**

Yes

**Position In Title:**

Yes

**Related Work:**

4

**Strengths And Weaknesses:**

Two different perspectives could exist on the contribution of this work. A significant amount of content is discussions of standard computer security practice (see S3 for example), and on those grounds one could argue about if this work provides anything new, besides trying to emphasise those points to the AI community. Especially when those who would be pushing for these kind of development practices would most likely not be strict AI people (who may be unaware of these concepts), but rather computer security people working for AI companies. The other perspective would be that general security knowledge in the AI community is limited (although this is a supposition, and has not necessarily been borne out by evidence), and a back to basics first principles discussion would provide interesting content to them. But would people from the latter group be interested in discussion based upon this work?

While I think this work is interesting, and broadly well written, my personal opinion is that I struggle to see this as necessarily being a position paper. Rather, I believe that this work contains strong foundations, that could be used for a more human-user style study of bug disclosure and management for users of AI systems, or a quantitative study of behaviours and practices in AI professionals. Take for example S4, and the checklist for third-party AI-evaluators, and the Figure 3 Flaw Report Card Schema - to me, this would be infinitely more interesting if it was based upon real user experiments and data, rather than just being a proposed system. It's easy to write a report card schema, the difficult thing is verifying its utility, and gauging how the set up of such a thing affects buy in. Fundamentally, without a pathway towards adoption and buy-in, I don't see how a work like this moves the needle in any way, and as such I don't see it provoking discussion within the community, or leading to changes in overall practice.

My overall tentative review score is based upon both my personal opinion, and an acknowledgement that there could be some pockets in the community that could benefit from reading this work. With all that said though, I must commend the authors on a well presented structure, with both narrative progression and writing that supports their arguments, which are well backed up by a modern and relevant view of the literature of AI security.

I will note though that I am not a fan of the appendices, and specifically, the choice to put certain content within the appendices. For example, "Key Definitions", and the majority of Appendix C, read to me as core content. Supplementary materials should go in the appendices, but this does not read as supplementary in the slightest, and to me the placement of this content seems to be an end-run around the page limit.

Other notes:
- L30 Missing space before "Second". Also, this probably should be "Secondly". Sentence also discusses "system providers" without making it clear who these system providers are, how they operate - treating structural perspectives as assumed knowledge.
- L46: No clear definition of GPAI systems.
- L28C2: Clarity regarding the processes may be useful - see for example, the pre-deployment stage.
- Figure 2: First-Third party is explained, but limited/expansive isn't. Also, the way that this is presented suggests a gradiation, but what is the gradiation that has first party limited having the greater security, rather than first party expansive? There may be a justification here, but it is not well explained, and the figure structure itself may be driving these issues. Also in text, it says "Our analysis focuses third-party AI evaluators" - but the authors do not clarify how the existence of first and second party approaches interact with their considerations.
- L129-144: The authors present their literature review with so little context, that it's almost meaningless. Yes, a literature review / overview should not be going into detail, but similarly, it should not be required that a reader goes through multiple citations in order to understand the meaning behind a paragraph. In L129-144, you talk about risks being "opaque....and can be difficult to foresee" - besides the structural repetition in the allusions, none of this tells the reader about where these risks are coming from. A reader unfamiliar with this space would not be able to take anything away from this content.
- Figure 3: In text, this is listed as being derived from a number of extant templates. But how was it derived from this? What does it gain / differ from different templates? Based upon the presentation, it is impossible to tell if 99% of the content is drawn from prior works, or 1%.
- App B.3 - the tie in with specific developers and systems intrinsically dates something like this (also, the formatting is not to the standard of the rest of your work). Surely the value would be in presenting a general framework, rather than tying the system to specific things.

**Support:**

2

---

> ### Author Rebuttal · Authors · 2025-03-29
>
> We would like to thank Reviewer Scur for their constructive feedback and thoughtful critiques. We are encouraged that they found our arguments well supported by a modern view of AI security, and that these arguments ground our position.
>
> Reviewer Scur provides an incredible amount of helpful feedback, which we largely agree with, and as such we commit to making specific edits to our paper, as detailed below. But primarily, in this rebuttal, we hope to address their core concern, that the work wouldn’t provoke discussion in the community or pave a feasible path for changes in practice.
>
> **How would this work provoke discussion within the community?**
>
> First, we would like to convince you that our position is not well accepted in the AI community—and even somewhat controversial. While the average ICML attendee might agree standardized flaw reporting is beneficial, we have found AI researchers often disagree that broad third-party evaluations (a) are a priority, (b) can be incentivised or protected without enabling malicious research, or (c) need to be coordinated across providers, or with the public. Note that no AI developer currently offers safe harbors or comprehensively includes flaws in its disclosure program, suggesting that the most powerful actors in the AI ecosystem largely don’t fully agree with these positions. And ML experts working with governments have largely not been successful in persuading key bodies such as the US Copyright Office that the near term risks of additional white-hat hacking are outweighed by the long term benefits of legal protections for good faith *safety* research. We have encountered many AI researchers that are broadly unaware of cybersecurity practices or skeptical of their transferability to AI.
>
> For these reasons, we don’t believe there is evidence these ideas have been broadly accepted by the AI community, and believe they would foster important discussion. We acknowledge the reviewers’ valid concerns, and commit to strengthening the work by emphasizing these points in the discussion and elsewhere the reviewer suggests.
>
> **Provide definitions and their context earlier.**
>
> We will move the definitions up-front, by transplanting and contextualizing the Key Definitions from the appendix. Further to your feedback, we will also take strides in the next draft to better contextualize our sources, explaining their main claims/evidence directly, in addition to citing them in-line.
>
> **Figure 2: Needs revisiting—the gradation, limited/expansive labels, and relationship between first, second, and third-party need greater explanation.**
>
> This is great feedback. We plan to switch this Figure to the appendix in exchange for the Key Definitions, as this figure is not a focus of our position. We will clarify the gradation in the figure, and the relationship between these evaluation modes: mainly that they are not just substitutes for one another, but complementary, and often target different flaws.
>
> **Figure 3: How was it derived, and from what?**
>
> Great question—we will expand on how the report template was derived. We synthesized and prioritized fields with external experts, and from forms used by the AIVD, MITRE, OECD, OpenAI, and CERT. Our broader contributions are shown in appendix Fig A6 and Table A1, where the form expands depending on certain criteria. We worked to balance collecting extensive information with simplicity/ease-of-use. We agree with the reviewer and will expand on our documentation of the design process and decisions.
>
>
> **Question: What do you foresee as the outcome of this work?**
>
> There are a few concrete outcomes we hope for and are working towards.
> * First, we hope the community will debate the proposed reporting schema, rules of engagement, and safe harbor templates.
> * Second, following this feedback, we hope to build and open source a simple reporting form. A successful form should enable quick reoprting, routing, and triaging.
> * Third, we hope AI providers will see the value and adopt AI flaw disclosure programs, along with safe harbors, and coordination.
>
> **Question: What would be the path to adoption of this work?**
>
> As you suggest in the latter part of your question, we plan to adapt extant systems and infrastructure from security, bringing it to AI. Please see our response to Reviewer nbqs, who shared your question. We specify a path to adoption for each component of our proposal.
>
> **Question: Would there be any negative consequences of corporatising and incentivising AI security research?**
>
> Yes, that’s very feasible—but in our opinion,more so with corporatising than incentivising AI security research! We’d be happy to discuss more here or in the text, if you have more context?
>
>
> Finally, we’d like to thank you again for your detailed and constructive feedback. We believe and hope these changes address your core critiques. Otherwise, please let us know what other changes we can make to improve the standing of this work in your review. Thank you!

---

> > ### Comment · Reviewer_Scur · 2025-04-01
> >
> > Thank you for your detailed and considered response. I'll come straight to the point - if there was a 3.499999 as a score, that's where I'd be at after your response. So for the AC's benefit, I'm very weak on the weak part of weak accept. I will also actively think on this point, and I may change my score after further contemplation.
> >
> > My one caveat, and, unfortunately, this is still kind of a major one relates to the statement "we have found AI researchers often disagree that broad third-party evaluations (a) are a priority, (b) can be incentivised or protected without enabling malicious research, or (c) need to be coordinated across providers, or with the public". How have you found this? This goes back to the point in my initial review that I believe this work would be well suited to quantitative study - using interviews, surveys, or anything else, to help support the arguments that you are making. I know this would be somewhat tangential to the main point the authors are trying to put forward, but I also do believe that this is one of the foundations to your overall argument, and as a foundation it's a little soft and crumbly. This is all not to say that I think it's a fatal flaw, but I do find it notable.

---

> > > ### Author Response · Authors · 2025-04-02
> > >
> > > Thank you as well! We appreciate this feedback and are encouraged by your positive assessment of our proposed edits!
> > >
> > > What if we ground the evidence for a lack of support in coordinated disclosure directly in empirical metrics—e.g. a table measuring adoption by major AI providers? This would support our argument with an empirical foundation, as you recommend. Something like:
> > >
> > > * For 10-20 prominent AI providers: e.g. Google, OpenAI, Anthropic, Meta, Cohere,...
> > > * How many have any issue reporting/feedback forms?
> > > * How many have dedicated disclosure programs? Are they invite-only?
> > > * How many have some form of flaw sharing/coordination commitments (eg the recent FMF announcement)?
> > > * How many have safe harbors? Or safe harbors that meet criteria listed in the paper?
> > > * How many have *invited* third-party research programs?
> > >
> > > We’ve already done most of this analysis for 5 developers in Table A.3. It wouldn’t be burdensome to extend it to several more, and succinctly measure/rank these levels of adoption (of third-party eval/disclosure programs, coordination, and legal protections) in the main paper.
> > >
> > > Let us know if this would help address most of your caveat? We appreciate your engagement and feedback!

---

### Official Review · Reviewer_b91c · 2025-03-14

**Significance:** 4
**Argument Clarity:** 3
**Rating:** 4
**Confidence:** 5

**Questions:**

One problem in actually implementing this disclosure pipeline is that how to coordinate various stakeholders so that they all adopted the proposed framework?

**Discussion Potential:**

4

**Paper Summary:**

This paper advocates for a new process of coordinate flaw reports across many stakeholders in general purpose AI systems.
The position is inspired by the flaw reports infrastructure in software security and proposed a new flaw reporting system for GPAI systems including AI Flaw Report Card Schema and Legal safe harbor rules.
Alternative views on AI evaluations are discussed.

**Position:**

Yes

**Position In Title:**

Yes

**Related Work:**

4

**Strengths And Weaknesses:**

Strength,

The proposed position for a new GPAI flaw reporting system seems well supported with the long standing practice in the software security domain, and detailed templates and schema are also proposed for the community to adopt.
The paper makes a very valid argument on this new paradigm of AI flaw disclosure pipeline.

Weakness,

I believe this position gives a very well described point of view towards AI flaw reports and I did not find obvious weaknesses from this paper.

**Support:**

4

---

> ### Author Rebuttal · Authors · 2025-03-29
>
> We would like to thank Reviewer b91c for their constructive feedback and their strong support of our submission! We appreciate that they recognize the work we put into the detailed templates and schema. We also appreciate that they recognize the imperative of transferring the lessons from the software security domain to AI, where the practices remain largely unadopted.
>
> **Question: How would one implement the disclosure pipeline to coordinate various stakeholders, so that they all adopt the proposed framework?**
>
> This is a great question, and one that we only begin to answer. In short, we designed the flaw report schema to be maximally interoperable with all the stakeholders that may want to receive them. We designed the schema to contain all the recommended fields from security vulnerability reporting, and from the AI flaw reports that do exist (eg. from DEFCON 2024 [1]), as well as the early reporting forms from OpenAI and other developers. Our goal is to make it convenient for developers, researchers, and other stakeholders to easily generate and consume reports.
>
> We are already starting to see major developers create nascent versions of these pipelines (e.g. Anthropic’s invite-only Bug Bounty), and so are optimistic that this work is helping promote broader interoperability and coordination as the ecosystem starts to head in this direction. We hope that by outlining the need for a pipeline for coordinated flaw disclosure and providing usable templates, we will spark/accelerate adoption, and discussion as to the best way of making coordinated flaw disclosure a reality.
>
> [1] McGregor et al (2024), To Err is AI: A Case Study Informing LLM Flaw Reporting Practices. https://arxiv.org/abs/2410.12104, IAAI 2024.

---

### Official Review · Reviewer_nbqs · 2025-03-25

**Significance:** 4
**Argument Clarity:** 4
**Rating:** 4
**Confidence:** 4

**Questions:**

1. Who could be possible parties who would retaliate to your suggestions? How would you reason with said parties?
2. The systems and the governing bodies recommended were easy to envision but in reality, would they be part of CISA, NIST, or a nonprofit?
3. How would you propose the governance of the “Disclosure Coordination Center”?

**Discussion Potential:**

3

**Paper Summary:**

The paper believes industries like software security are much ahead in reporting system flaws and calls for better infrastructure and practices to report flaws in General Purpose AI (GPAI) systems. The authors feel third-party risk evaluations have specific benefits in particular as they enhance scalability, the coverage of evaluations, and evaluator independence. The authors propose 3 recommendations to  boost safety and security of GPAI systems:
1. Third-party evaluators should submit AI flaw reports
and abide by standardized rules of conduct.
2. GPAI system providers should adopt flaw disclosure
programs with safe harbors for third-party
evaluation.
3. Providers and evaluators should partner to establish
coordinated flaw disclosure.

They support their recommendation with a report template, and standardized rules of conduct adapted from the operationalization of “good-faith research” in computer security. This paper essentially calls for a security-style ecosystem for GPAI that is open, collaborative, and legally safe for good-faith research to ensure scalable and effective safety oversight beyond the developers themselves.

**Position:**

Yes

**Position In Title:**

Yes

**Related Work:**

4

**Strengths And Weaknesses:**

Strengths:
1. The paper states its position clearly and supports the position with relevant prior work and good suggestions to mitigate the identified issue of lack of a standardized system to report GPAI software flaws.
2. The paper explicitly addresses six well chosen common misconceptions that serve as counterarguments to its position, with logical rebuttals.
3. The paper has strong grounding in software security with relevant analogies and instances.
4. The AI report card was a sensible addition that will help GPAI system stakeholders visualize (and later implement) the authors' recommendation.

Weaknesses:
1. In addition to the analogies present, it would be nice to have chosen case studies where there was a noticeable amelioration in the problem when a solution similar to the recommendations provided was implemented.
2. It is unclear to me who (or what organization) would have the power to establish the norms suggested (like the “Safe Harbor legal templates”). If a separate body is formed for implementing the measure, it would be nice if the authors provided a functioning scenario for the same (not unlike the template they provided).

**Support:**

3

---

> ### Author Rebuttal · Authors · 2025-03-29
>
> We are very encouraged by Reviewer nbqs’s positive feedback, and recognition that this work presents a clear and well supported position. We particularly appreciate the recognition that our work explicitly addresses common misconceptions and counterarguments, and provides new designs for AI flaw reporting. We answer the reviewer’s suggestions below, which we will incorporate into the paper.
>
> **Discuss case studies in more depth, demonstrating where our recommendations have helped other industries.**
>
> We will add more details directly to the paper, including:
>
> At DEFCON 2024, AI2’s OLMo and WildGuard were subjected to 495 third-party hackers [1]. Red teamers submitted candidate flaws through structured reporting mechanisms. The session yielded the successful identification, triaging, and mitigation of several notable flaws: a templated jailbreak to induce harmful responses at 20-50% rates, a prompting strategy to generate legal misinformation, and the model’s propensity to comply with harmful conspiracies when they were presented as factual in the input. These flaws were used to improve development of AI2’s system and were reflected in edited system cards.
>
> Additionally, the 2022 Defense Industrial Base Vulnerability Disclosure Program (DIB-VDP) Pilot serves as a successful application of coordinated cybersecurity flaw disclosure [2]. It led to the discovery of 403 vulnerabilities, whose remediation saved different companies that are part of the Defense Industrial Base an estimated $61.4 million dollars by preventing cyberattacks.
>
>
> **What organizations could establish the suggested norms? (And your second and third questions about governing bodies.)**
>
> This is a great question, and one we agree we should dedicate more space to. In short, several organizations are positioned to aid in flaw coordination, including NIST, the Frontier Model Forum, AI Alliance, OECD, CERT, MITRE, CISA, and major developers and deployers. We will add a subsection to the paper to address these possibilities, expanding on the details below.
>
> Our recommendations are not designed to have one governing body or significant adoption effort. Ideally, an array of actors can adopt them for their own uses. Here we highlight how adoption might work for our recommendations:
> * **Standardized AI flaw reports:** open sourcing these tools would allow any major AI developer to host their own version of the form. Also, researchers and red teamers could fill out the schema themselves, to make flaw reporting more convenient and useful to those they send the reports to.
> * **Disclosure programs and safe harbors:** These are voluntary for each AI system provider. For example, already Anthropic has partnered with HackerOne to establish an initial Model Safety Bug Bounty. Regulators may also mandate disclosure programs and safe harbors, as anticipated by the EU AI Act’s Code of Practice Process for General-Purpose AI.
> * **Disclosure coordination:** Organizations like CERT or MITRE may be best positioned to coordinate flaw disclosure based on their existing work in this area and close collaboration with government agencies like CISA. Our analysis shows that they also already provide this functionality for security vulnerabilities and hope to do so for AI as well, but have seen limited adoption as many AI flaws are emailed to one developer rather than sent to these organizations.
>
> **Question: What parties would disagree with your suggestions, and how could we reason with them?**
>
> Generally, we’ve found a lack of consensus on the value of third-party evaluations and disclosure coordination altogether, within the AI community. In our discussions with experts, some believe testing is best left to internal product teams, or that flaw awareness should not extend beyond the affected system. We believe that these views have been shown counter-productive in practice and literature (our Section 3) for software security, and this is also the case for AI systems. We hope to convince them by presenting the evidence from DEFCON [1], from new bug bounties (such as Anthropic’s recent one), and the empirically verifiable benefits to software security (see Misconception 1).
>
> Large AI developers are also skeptical: they do not currently have safe harbors for third-party researchers or encourage standardized flaw disclosure, both because they are legally cautious and concerned about inviting additional white-hat hacking and sometimes because they do not wish to acknowledge flaws in their systems. We hope that by building and adopting these tools we can convince companies with empirical evidence that they improve AI safety and security.
>
>
> [1] McGregor et al (2024), To Err is AI: A Case Study Informing LLM Flaw Reporting Practices. https://arxiv.org/abs/2410.12104, IAAI 2024.
>
> [2] DoD Cyber Crime Center. Vulnerability disclosure program annual report 2022, 2022. https://www.dc3.mil/Portals/100/Documents/DC3/Missions/VDP/Annual%20Reports/2022/VDP-2022-Annual-Report-Final.pdf

---

### Decision · Program_Chairs · 2025-04-30

**Decision:**

Accept (spotlight poster)

**Comment:**

This paper argues that third-party evaluation and flaw disclosure are critical for GPAI safety, borrowing lessons from the software security world. The position is clear, the recommendations are concrete (flaw reporting templates, safe harbor guidance, disclosure coordination), and the authors directly engage with pushback.

Multiple reviewers were positive. Reviewer b91c said, “The paper makes a very valid argument… I did not find obvious weaknesses.” Reviewer nbqs highlighted the “strong grounding in software security” and appreciated that “the paper explicitly addresses six well-chosen misconceptions.” Reviewer Scur raised a fair concern about whether the community truly lacks support for these ideas, but ultimately bumped their score after the authors committed to adding empirical grounding (e.g., adoption tables): “I will raise my score to a 4… assuming a variant of A.3 would be prepared.”